# Developmental 'awakening' of primary motor cortex to the sensory consequences of movement

James C Dooley[1,2]*, Mark S Blumberg[1,2,3,4,5]

[1]Department of Psychological & Brain Sciences, University of Iowa, Iowa, United States; [2]DeLTA Center, University of Iowa, Iowa, United States; [3]Interdisciplinary Graduate Program in Neuroscience, University of Iowa, Iowa, United States; [4]Department of Biology, University of Iowa, Iowa, United States; [5]Iowa Neuroscience Institute, University of Iowa, Iowa, United States

**Abstract** Before primary motor cortex (M1) develops its motor functions, it functions like a somatosensory area. Here, by recording from neurons in the forelimb representation of M1 in postnatal day (P) 8–12 rats, we demonstrate a rapid shift in its sensory responses. At P8-10, M1 neurons respond overwhelmingly to feedback from sleep-related twitches of the forelimb, but the same neurons do not respond to wake-related movements. By P12, M1 neurons suddenly respond to wake movements, a transition that results from opening the sensory gate in the external cuneate nucleus. Also at P12, fewer M1 neurons respond to individual twitches, but the full complement of twitch-related feedback observed at P8 is unmasked through local disinhibition. Finally, through P12, M1 sensory responses originate in the deep thalamorecipient layers, not primary somatosensory cortex. These findings demonstrate that M1 initially establishes a sensory framework upon which its later-emerging role in motor control is built.
DOI: https://doi.org/10.7554/eLife.41841.001

*For correspondence:
james-c-dooley@uiowa.edu

Competing interests: The authors declare that no competing interests exist.

## Introduction

In placental mammals, primary motor cortex (M1) plays a critical role in adapting behavior to an ever-changing environment (*Kawai et al., 2015*). Interestingly, M1 does not assume this role until relatively late in development (*Chakrabarty and Martin, 2000*; *Flament et al., 1992*; *Martin et al., 2005*; *Müller et al., 1991*; *Nezu et al., 1997*; *Olivier et al., 1997*; *Young et al., 2012*). For example, in rats, intracortical microstimulation of M1 neurons does not evoke movements until postnatal day (P) 35 (*Young et al., 2012*). It is not understood why M1 shows such protracted development or how M1 functions before it assumes its 'motor identity.' One possibility is that M1 first develops a sensory framework, and it is upon this framework that its later-emerging motor functions rest (*Bruce and Tatton, 1980*; *Chakrabarty and Martin, 2005*).

Beginning early in development in rats, M1 neurons respond to externally generated (i.e. exafferent) stimulation (*An et al., 2014*; *Asanuma, 1981*; *Tiriac and Blumberg, 2016*). In addition, M1 neurons respond to sensory feedback (i.e. reafference) arising from myoclonic twitches—the discrete, jerky movements produced abundantly and exclusively during active (REM) sleep (*Tiriac et al., 2014*). With respect to reafference from wake movements, robust M1 responses are readily observed in adults (*Asanuma, 1981*; *Fetz et al., 1980*; *Georgopoulos et al., 1982*), but not in infants at 1 week of age (*Tiriac et al., 2014*). Specifically, in infants, wake-related reafference in the forelimb representation of M1 is suppressed, due to selective inhibition within the external cuneate nucleus (ECN), a medullary nucleus that receives primary proprioceptive afferents from forelimb muscle spindles and conveys that afferent information to downstream structures (*Boivie and Boman,*

*1981*; *Campbell et al., 1974*); disinhibiting the infant ECN unmasks wake-related reafference (*Tiriac and Blumberg, 2016*). Consequently, M1 appears to be 'blind' to the proprioceptive consequences of wake movements early in development.

In adults, proprioceptive input to M1 arrives both directly from thalamus and indirectly through primary somatosensory cortex (S1; *Asanuma, 1981*). Contemporary theories of M1's role in motor learning assume it has access to this proprioceptive input, permitting continuous monitoring of limb position and updating of motor commands (*Mathis et al., 2017*). These theories take for granted both M1's role in producing movements and the availability of reafference from those movements, neither of which pertains to M1 early in development. However, as development progresses and M1 assumes its motor functions, there must come a time when M1 neurons begin to respond to proprioceptive feedback from wake movements; it is not known when that occurs. Interestingly, although intracortical microstimulation of M1 neurons does not normally evoke movement until P35, movements can be evoked at P13—but not at P12—after local disinhibition with the $GABA_A$ antagonist, bicuculline (*Young et al., 2012*). This suggests that as late as P12, corticospinal neurons are unable to directly drive movement, which is consistent with the expansion of corticospinal axons in the spinal cord observed around this time (*Curfs et al., 1994*; *Schreyer and Jones, 1982*).

By performing extracellular recordings in the forelimb representation of M1 in unanesthetized P8-12 rats as they cycled between sleep and wake, we first show that M1 neurons at P8-10 are exclusively responsive to reafference from twitches, consistent with previous studies (*Tiriac and Blumberg, 2016*; *Tiriac et al., 2014*). Then, suddenly between P10 and P12, this pattern changes such that M1 neurons now respond to reafference from wake movements, but less often to twitches. Because M1 neurons at all ages respond to exafferent stimulation, this developmental switch must be due specifically to changes in the processing of sensory input arising from self-generated movements. To that end, we identify two separate developmental events around P12: (i) an upstream change in the gating of wake-related reafference in the ECN and (ii) a local suppression of twitch-related reafference within M1. Together, these events comprise a rapid shift in M1 sensory processing that constitutes an 'awakening' to the sensory consequences of movement just before the developmental emergence of motor outflow.

## Results

We recorded neural activity from the forelimb representation of M1 in head-fixed, unanesthetized rats between P8 and P12 (P8: n = 10 pups, 160 neurons; P9: n = 9 pups, 112 neurons; P10: n = 8 pups, 123 neurons; P11: n = 15 pups, 197 neurons; P12 pups: n = 7, 165 neurons) using 4-shank

**Table 1.** Details for all groups of infant subjects in this study.

| Expt. | Age | # of Animals | Mean Weight (g) | # of Neurons | Neuronal Classification | | | | | Mean # of Twitches | Mean # of Wake Movements |
| | | | | | Unresp. | Resp. | Wake | Twitch | Both | | |
|---|---|---|---|---|---|---|---|---|---|---|---|
| M1 | 8 | 10 | 19.0 ± 0.40 | 160 | 84 | 76 | 0 | 64 | 12 | 302 ± 31 | 60 ± 11 |
| | 9 | 9 | 21.8 ± 0.60 | 112 | 61 | 51 | 0 | 41 | 10 | 259 ± 10 | 72 ± 10 |
| | 10 | 8 | 24.8 ± 0.27 | 123 | 73 | 50 | 10 | 26 | 14 | 187 ± 26 | 54 ± 10 |
| | 11 | 15 | 27.4 ± 0.34 | 197 | 113 | 84 | 23 | 18 | 43 | 157 ± 18 | 79 ± 10 |
| | 12 | 7 | 30.2 ± 0.74 | 165 | 102 | 63 | 48 | 3 | 12 | 104 ± 22 | 82 ± 14 |
| ECN | 9 | 6 | 20.3 ± 1.18 | 16 | 0 | 16 | 1 | 11 | 4 | 253 ± 20 | 84 ± 15 |
| | 12 | 7 | 28.1 ± 1.20 | 20 | 0 | 20 | 1 | 4 | 15 | 226 ± 27 | 70 ± 8 |
| Saline Pre | 12 | 6 | 31.1 ± 1.27 | 107 | 73 | 38 | 30 | 2 | 6 | 177 ± 43 | 113 ± 8 |
| Saline Post | | | | | 69 | 34 | 25 | 3 | 6 | 186 ± 29 | 97 ± 14 |
| Bicuc. Pre | 12 | 6 | 30.0 ± 1.23 | 99 | 56 | 43 | 39 | 2 | 2 | 112 ± 19 | 80 ± 13 |
| Bicuc. Post | | | | | 34 | 65 | 19 | 7 | 39 | 131 ± 37 | 91 ± 19 |

The number of animals, weight (mean ± s.d.), neuronal classification, and number of triggered twitches and wake movements (mean ± s.d.) for the M1, ECN, and disinhibition experiments.

DOI: https://doi.org/10.7554/eLife.41841.002

silicon depth electrodes (*Figure 1—figure supplement 1a,c*; *Table 1*). Neuronal activity, as well as nuchal, forelimb, and hindlimb electromyographic (EMG) activity, was recorded for 30 min as pups cycled between sleep and wake (*Figure 1a*). We confirmed electrode placement in the forelimb representation of M1 through exafferent stimulation of the forelimb as well as subsequent staining of flattened, tangentially sectioned brains for cytochrome oxidase (CO; *Figure 1b*). M1 recording sites were restricted to agranular cortex immediately medial to S1 (*Figure 1c–d*). In a subset of coronally sectioned brains, recording depth was confirmed to be restricted to the deeper layers of cortex (*Figure 1—figure supplement 1a–c*).

## Rapid developmental onset of sensory responsiveness in M1 neurons

As reported previously in P8 rats (*Tiriac and Blumberg, 2016*; *Tiriac et al., 2014*), neurons in the forelimb representation of M1 exhibited more activity during periods of active sleep than during periods of wake (*Figure 1e*, left). Perievent histograms revealed that neuronal activity in M1 during active sleep clustered around forelimb twitches, but not wake movements (*Figure 1e*, right). The pattern of neural activity at P9 and P10 largely resembled that at P8. However, at P11, M1 neurons were substantially more active during wake (*Figure 1f*, left), with increased responsiveness to wake movements in addition to continued responsiveness to twitches (*Figure 1f*, right). By P12, M1 neurons were continuously active during sleep and wake (*Figure 1g*, left). Similar to P11, M1 activity increased after forelimb wake movements at P12, but no longer increased after twitches (*Figure 1g*, right). Notably, of the 757 M1 neurons from which we recorded, not one showed a pattern of activity reflective of motor output—that is, an activity peak preceding movement (*Del Rio-Bermudez et al., 2015*)—thus providing the strongest evidence to date that M1 does not produce movement at these ages (see *An et al., 2014*).

Between P8 and P12, we observed an age-related decrease in the number of forelimb twitches (*Figure 1—figure supplement 2a,* $F_{(4,44)} = 11.4$, p<0.0001), consistent with previous reports (*Marcano-Reik et al., 2010*), but not in the number wake movements ($F_{(4,44)} = 1.1$, p=0.37). Importantly, because the EMG activity associated with twitches and wake movements did not change with age (*Figure 1—figure supplement 2b*), any age-related changes in the reafferent responses of M1 neurons to twitches and wake movements cannot be attributed to changes in the duration of these movements.

## Developmental shift in the population characteristics of neuronal responses in M1

To determine whether individual neurons were twitch- or wake-responsive, perievent histograms triggered on twitches or wake movements were constructed for each isolated M1 neuron. We then fit the histograms to models of idealized M1 neuronal responses using custom-written MATLAB code. Twitches, being discrete movements, were fit to a symmetrical Gaussian function (*Figure 2a*) and wake movements were fit to an asymmetrical function comprising a Gaussian with an exponential decay (*Figure 2b*). We used these models to perform regression analyses on each neuron, assigning each to a response category based on its adjusted $r^2$ values. The adjustment to the $r^2$ value was used to account for the number of regression coefficients (see Materials and methods), permitting a direct comparison of the fits for twitches and wake movements. Because $r^2_{adj}$ describes how much of the variance in a neuron's activity can be explained by the model, it functions as a responsivity index. (Hereafter, all reported $r^2$s are adjusted values.)

The threshold for distinguishing between responsive and unresponsive neurons was based on the mean change in firing rate (in relation to baseline) across neuron classifications. A threshold $r^2$ of 0.35 was selected because it most effectively differentiated neurons that responded to twitches and wake movements from those that did not (*Figure 3—figure supplement 1a*). Using this threshold, we classified all M1 neurons as either unresponsive ($r^2 \leq 0.35$ for both twitches and wake movements) or responsive ($r^2 > 0.35$ for twitches and/or wake movements). A representative unresponsive neuron, as well as representative responsive neurons from each of the three possible classifications (twitch-responsive, wake-responsive, twitch- and wake-responsive) are shown in *Figure 2c* . Mean perievent histograms of each neural classification can be found in *Figure 3—figure supplement 1a*. There was no age-related difference in the percentage of neurons classified as unresponsive ($\chi^2_4$= 5.3, p = 0.26; *Figure 3—figure supplement 1b*).

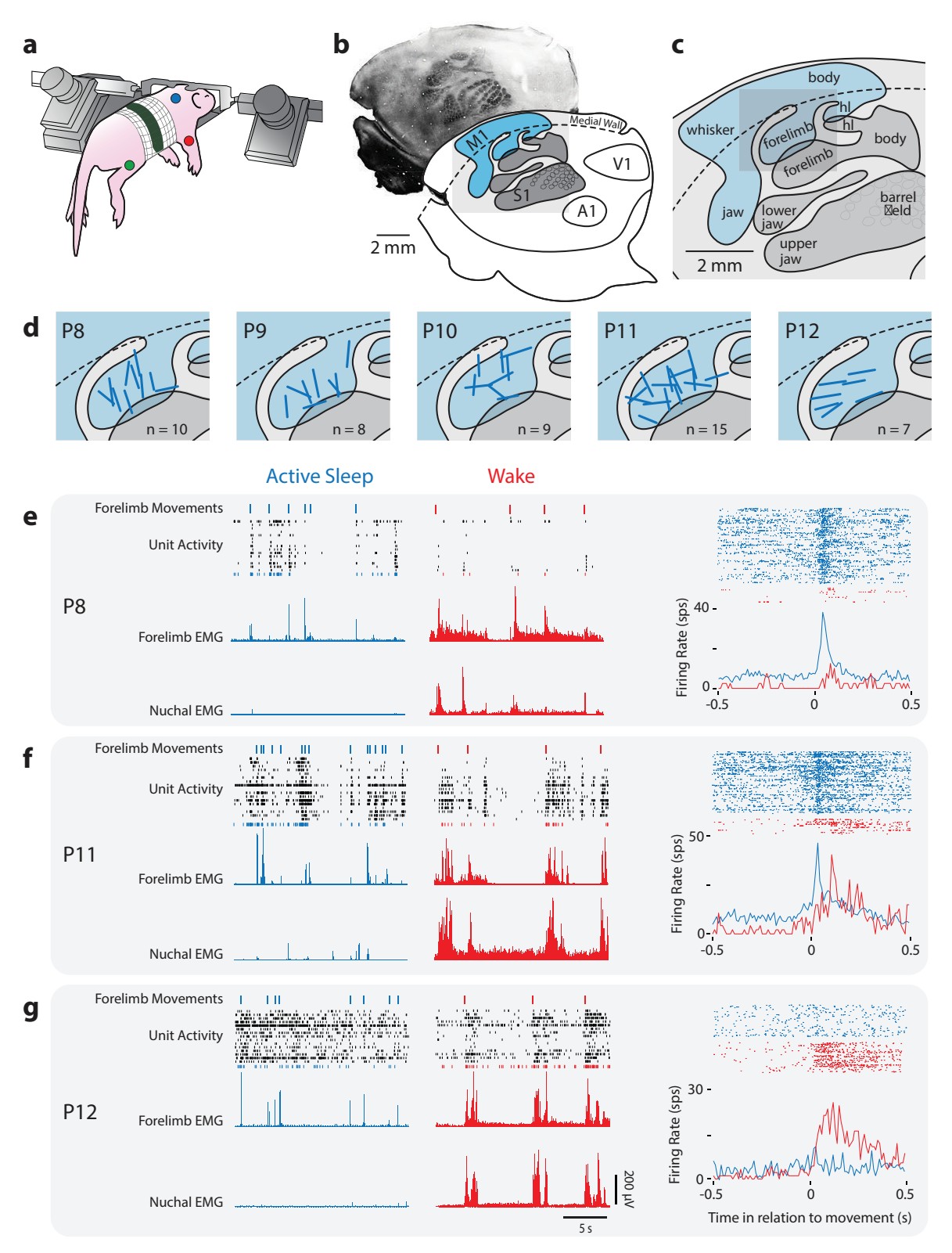

**Figure 1.** Rapid developmental transition in M1 sensory responsiveness. (a) The method used to record electrophysiologically from a head-fixed pup. Dots denote locations of the EMGs (forelimb, red; hindlimb, green; nuchal muscle, blue). (b) Top: Flattened cortex sectioned tangentially to the surface and stained for cytochrome oxidase (CO); primary somatosensory cortex (S1) appears darker than the surrounding tissue. Bottom: Boundaries of primary sensory areas from CO-stained tissue, illustrating S1 and primary motor cortex (M1), as well as primary auditory (A1) and visual (V1) cortex. (c)
*Figure 1 continued on next page*

*Figure 1 continued*

Enlargement of gray box in (b) showing the somatotopic organization within S1 and M1. hl: hindlimb. (d) Enlargement of gray box in (c) showing the locations of recording sites (blue bars) within the forelimb representation of M1. (e) Left: Representative data at P8 depicting 20 s periods of active sleep (blue) and wake (red), showing forelimb movements (twitches: blue ticks; wake movements: red ticks), unit activity within the forelimb representation of M1, and rectified EMGs from contralateral forelimb and nuchal muscles. Each row of depicts unit activity for a different neuron. The bottom-most neuron (blue or red), is represented further at right. Right, top: Raster sweeps for an individual M1 neuron triggered on twitches (blue) and wake movements (red), with each row representing a different movement. Right, bottom: Perievent histograms (bin size = 10 ms) showing the unit's mean firing rate triggered on twitches (blue) or wake movements (red). (f) Same as in (e) except at P11. (g) Same as in (e) except at P12.

DOI: https://doi.org/10.7554/eLife.41841.003

The following figure supplements are available for figure 1:

**Figure supplement 1.** M1 recording locations in coronal perspective.

DOI: https://doi.org/10.7554/eLife.41841.004

**Figure supplement 2.** Frequency and kinematics of twitches and wake movements across age.

DOI: https://doi.org/10.7554/eLife.41841.005

Scatterplots of $r^2_{twitch}$ vs. $r^2_{wake}$ for each neuron show a developmental shift in the neural population responses to twitches and wake movements, particularly between P10 and P12 (*Figure 3a*). Whereas most responsive neurons at P8 and P9 (and, to a lesser extent, P10) occupy the twitch-responsive quadrant (top-left, blue), by P11 most of the responsive neurons have shifted to the twitch- and wake-responsive quadrant (top-right, purple). By P12 the population has shifted again, with most responsive neurons now occupying the wake-responsive quadrant (bottom-right, red). This population-level shift in responsivity is supported quantitatively, with responsive neurons showing a significant decrease in $r^2_{twitch}$ from P8 to P12, the largest decrease being between P11 and P12 (*Figure 3b*, blue bars; $H_{(4,321)}$ = 85.4, p<0.001). Correspondingly, $r^2_{wake}$ increased from P8 to P12 (*Figure 3b*, red bars; $H_{(4,321)}$ = 119.6, p<0.001). The age-related reversal in responsiveness is most visually apparent as a change in the percentage of responsive neurons that are twitch-responsive, twitch- and wake-responsive, and wake-responsive across age (*Figure 3c*). Importantly, all responsive neurons also responded to exafferent stimulation of the forelimb (*Figure 3—figure supplement 1a*; green lines), thus demonstrating consistency in the recorded neuron's receptive field across age.

The decreased percentage of twitch-responsive neurons observed between P8 and P12 is accompanied by decreased variability in those neurons that did respond to twitches. Specifically, twitch-responsive neurons exhibited increases in peak firing rate, narrower response tuning (half-width at half-height), and decreased latency across these ages, and all of these measures were accompanied by less variability across neurons (*Figure 3—figure supplement 2*; top; *Table 2*). Further, P11 and P12 twitch-responsive neurons responded more consistently to every twitch than did twitch-responsive neurons at P8-10. Gaussian-exponential fits of wake-responsive neurons did not show similar developmental trajectories (*Figure 3—figure supplement 2*; bottom; *Table 2*), most likely because wake movements are inherently more noisy and variable than twitches.

## M1 sensory responses originate in deep cortical layers

Sensory inputs to M1 in adults have two distinct origins: one arriving from S1 via horizontal connections to the superficial layers of M1 (*Mao et al., 2011*) and a second arriving directly from the thalamus to the deeper layers of M1 (*Hooks et al., 2013*). Despite numerous studies investigating the source and strength of sensory inputs to M1, it is not known whether these two inputs develop sequentially or simultaneously. To determine which M1 layers receive reafference at P8 and P12, we recorded M1 activity using 16-site linear electrodes (N = 2 at each age; *Figure 4a,c*). At both ages, the strongest M1 neural responses were found in layers 5b and 6 (*Figure 4b,d*; upper). Consistent with this, current source density analysis revealed a current sink in layers 5a and 5b (*Figure 4b,d*; lower). From these observations, we conclude that sensory input to M1 arrives directly from the thalamus.

Although anatomical evidence suggests that horizontal projections between S1 and M1 do not drive neural activity in the deeper layers before P12 (*Anastasiades and Butt, 2012*), we nonetheless assessed whether M1 reafference is conveyed via S1 by recording simultaneously from M1 and S1 at P8 (n = 6 pups, 94 M1 neurons, 91 S1 neurons) and P12 (n = 6 pups, 135 M1 neurons, 107 S1

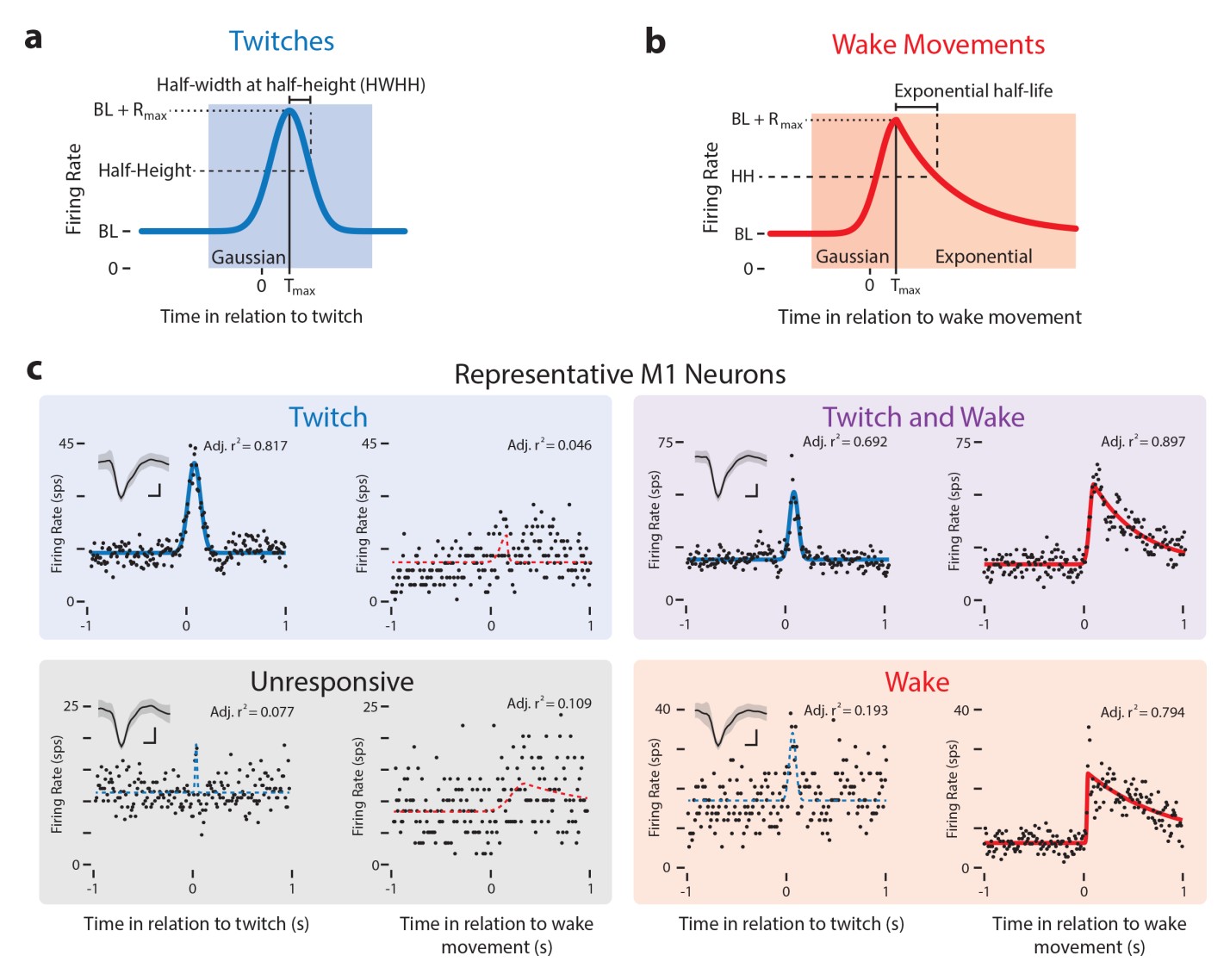

**Figure 2.** Modeling reafferent responses of M1 neurons to twitches and wake movements. (a) Gaussian function used to model M1 neural responses to twitches. Based on the model fits for twitch-triggered perievent histograms, we derived estimates of each neuron's baseline firing rate (BL), maximum response ($R_{max}$), peak time ($T_{max}$), and half-width at half-height (HWHH). (b) Gaussian-exponential function used to model M1 neural responses to wake movements. The function's rising phase is a Gaussian function, identical to that in (a). The falling phase is an exponential decay function. Parameters defined as in (a). (c) Representative perievent histograms and model fits illustrating all four neuron classifications: Twitch-responsive, unresponsive, twitch- and wake-responsive, and wake-responsive. Also shown is each neuron's Gaussian fit (blue lines) and Gaussian-exponential fit (red lines). Fits with adjusted $r^2$ values less than 0.35 are shown with a dotted line and fits with an adjusted $r^2$ value greater than 0.35 are shown with a solid line. Inset in the top left corner of is each neuron's mean waveform (black line) ±standard deviation. Vertical scale bar = 20 μV, Horizontal scale bar = 0.2 ms.
DOI: https://doi.org/10.7554/eLife.41841.006

neurons; *Figure 5a,b*). At both ages, we observed striking similarities between the activity profiles of responsive neurons in S1 and M1 to the same movements (*Figure 5c,d*). These similar responses were topographically restricted to forelimb twitches, as triggering perievent histograms on hindlimb or nuchal twitches did not yield strong responses. We then calculated cross-correlations of all M1-S1 pairs that were responsive to forelimb movement. To differentiate the portion of the cross-correlations due to a stimulus (e.g. twitch) from the component due to neuron-neuron interactions (e.g. horizontal projections), we analyzed the data using the shift predictor (*Engel et al., 1990*; *Perkel et al., 1967*). This analysis revealed that event-triggered S1 activity was typically contemporaneous with

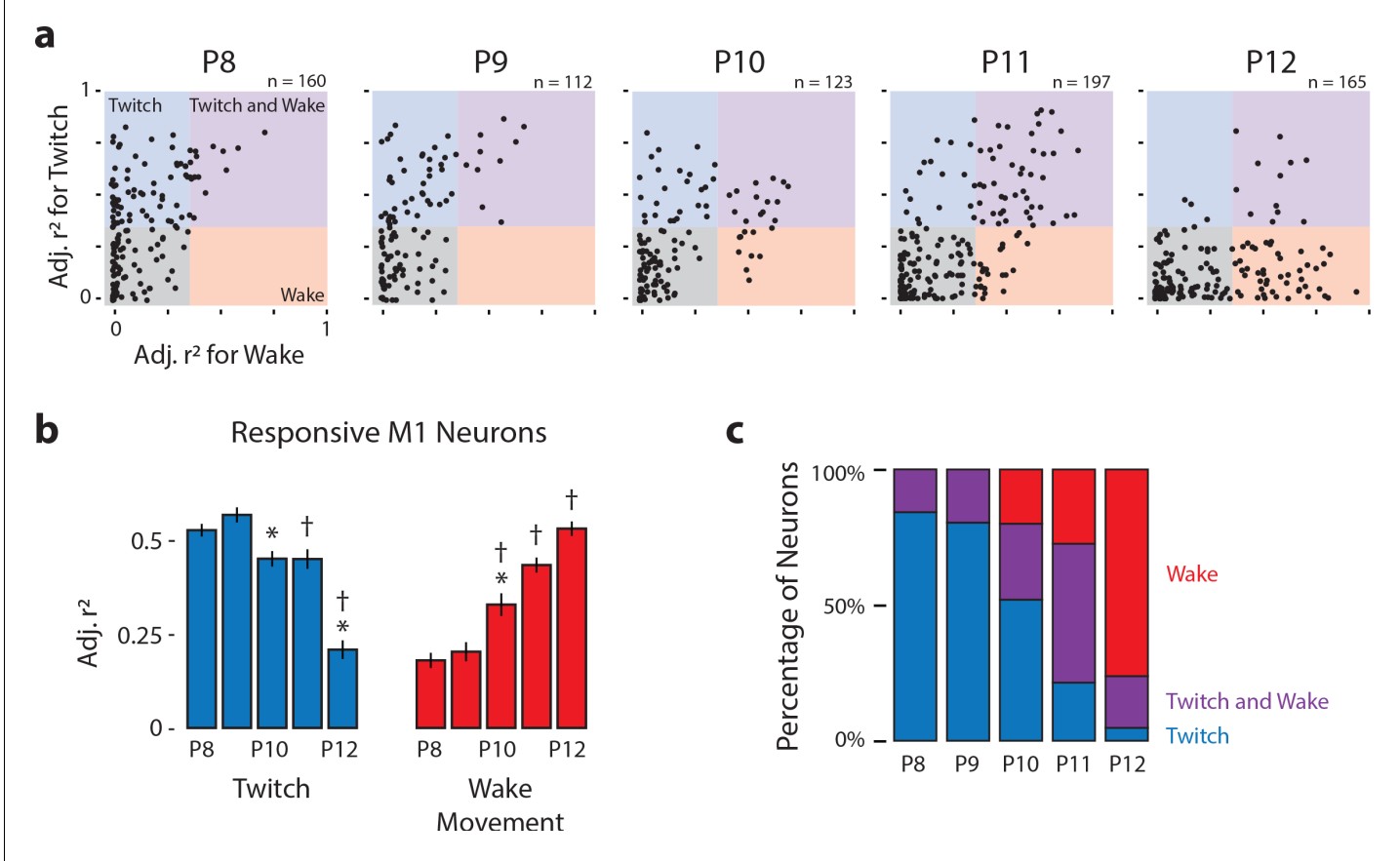

**Figure 3.** Developmental shift in the population characteristics of sensory responses of M1 neurons. (**a**) Scatterplot of each M1 neuron's adjusted $r^2$ value for twitches (y-axis) and wake movements (x-axis) across age. Background color illustrates the classification of neurons within that region of the scatterplot. (**b**) Mean (±SEM) adjusted $r^2$ values for twitches (blue) and wake movements (red) for responsive M1 neurons at each age. * Significant difference from previous day ($p<0.005$). † Significant difference from 2 days prior ($p<0.005$). (**c**) Percentage of neurons classified as wake-responsive (red), twitch- and wake-responsive (purple), and twitch-responsive (blue) at each age.

DOI: https://doi.org/10.7554/eLife.41841.007

The following figure supplements are available for figure 3:

**Figure supplement 1.** Mean perievent histograms by age, neuron response classification, and triggered event.

DOI: https://doi.org/10.7554/eLife.41841.008

**Figure supplement 2.** Developmental refinement of twitch-responsive neurons.

DOI: https://doi.org/10.7554/eLife.41841.009

M1 activity (*Figure 5c,d*), thus ruling out the conveyance of reafference from S1 to M1 via horizontal projections.

## Developmental onset of wake-related reafference in the ECN

At P8-10, the ECN acts as a sensory gate that selectively blocks wake-related reafference (*Tiriac and Blumberg, 2016*; *Figure 6a*). Thus, we hypothesized that the sudden emergence of wake movement responses at P11-12 reflects the opening of the ECN's sensory gate (*Figure 6b*). We tested this hypothesis by comparing ECN neuronal activity in unanesthetized, head-fixed rats at P8-9 (hereafter P9; n = 6 pups, 16 neurons) and P11-12 (hereafter P12; n = 7 pups, 20 neurons; *Figure 6—figure supplement 1a,b*; *Table 1*).

As described previously (*Tiriac and Blumberg, 2016*), we observed robust responses in P9 ECN neurons to forelimb twitches, but not wake movements (*Figure 6c*). At P12, however, ECN neurons responded to both twitches and wake movements (*Figure 6d*). Because the overall shapes of the perievent histograms for ECN and M1 neurons were similar, ECN neurons were also fit to Gaussian

**Table 2.** Model-fit parameters for M1 and ECN neurons.

**M1 Twitch-Responsive Neurons**

| Age | BL (sps) | $R_{max}$ (sps) | $T_{max}$ (s) | HWHH (s) | | % Responding |
|---|---|---|---|---|---|---|
| P8 | 2.12 (1.05, 3.66) | 3.86 (2.57, 7.65) | 0.194 (0.141, 0.252) | 0.186 (0.137, 0.220) | | 25.7 (18.1, 35.3) |
| P9 | 2.52 (1.67, 4.45) | 7.14 (4.72, 9.62) | 0.119 (0.095, 0.158) | 0.122 (0.086, 0.154) | | 31.5 (25.5, 38.8) |
| P10 | 2.87 (1.68, 3.86) | 5.18 (3.71, 9.94) | 0.172 (0.111, 0.229) | 0.124 (0.102, 0.192) | | 24.1 (19.0, 32.7) |
| P11 | 5.20 (3.18, 10.00) | 14.95 (9.13, 27.22) | 0.075 (0.054, 0.094) | 0.085 (0.053, 0.132) | | 36.4 (27.8, 50.7) |
| P12 | 6.45 (4.11, 12.37) | 12.82 (9.10, 20.10) | 0.068 (0.056, 0.095) | 0.105 (0.084, 0.115) | | 45.8 (25.6, 54.6) |

**M1 Wake-Responsive Neurons**

| Age | BL (sps) | $R_{max}$ (sps) | $T_{max}$ (s) | HWHH (s) | $\lambda$ (s) | % Responding |
|---|---|---|---|---|---|---|
| P8 | 0.63 (0.39, 0.96) | 6.70 (3.89, 8.93) | 0.199 (0.185, 0.265) | 0.178 (0.115, 0.229) | 0.231 (0.154, 0.352) | 57.7 (38.9, 64.7) |
| P9 | 1.04 (0.68, 1.55) | 7.33 (5.59, 9.25) | 0.109 (0.097, 0.120) | 0.065 (0.059, 0.086) | 0.298 (0.087, 0.364) | 56.6 (46.5, 67.1) |
| P10 | 1.23 (0.52, 1.53) | 9.62 (6.83, 13.14) | 0.214 (0.166, 0.235) | 0.091 (0.067, 0.116) | 0.213 (0.114, 0.368) | 60.4 (42.3, 68.5) |
| P11 | 2.63 (1.37, 5.35) | 10.14 (6.63, 15.85) | 0.095 (0.061, 0.165) | 0.079 (0.028, 0.137) | 0.401 (0.182, 1.319) | 68.0 (51.0, 79.5) |
| P12 | 2.31 (0.99, 4.64) | 9.05 (5.00, 14.10) | 0.126 (0.052, 0.260) | 0.089 (0.033, 0.239) | 0.323 (0.181, 0.539) | 60.6 (46.1, 69.3) |

**ECN Twitch-Responsive Neurons**

| Age | BL (sps) | $R_{max}$ (sps) | $T_{max}$ (s) | HWHH (s) | | % Responding |
|---|---|---|---|---|---|---|
| P9 | 5.97 (3.19, 9.73) | 14.59 (7.33, 24.79) | 0.043 (0.036, 0.051) | 0.063 (0.026, 0.085) | | 31.0 (17.1, 43.1) |
| P12 | 3.39 (2.33, 6.01) | 23.42 (8.22, 34.07) | 0.032 (0.027, 0.044) | 0.031 (0.023, 0.036) | | 41.7 (23.3, 52.4) |

**ECN Wake-Responsive Neurons**

| Age | BL (sps) | $R_{max}$ (sps) | $T_{max}$ (s) | HWHH (s) | $\lambda$ (s) | % Responding |
|---|---|---|---|---|---|---|
| P12 | 3.66 (1.61, 6.13) | 19.94 (11.13, 35.38) | 0.054 (0.038, 0.074) | 0.058 (0.037, 0.079) | 0.166 (0.034, 0.428) | 74.1 (59.4, 85.9) |

The median values (along with the 25th and 75th percentiles) for all twitch-responsive and wake-responsive neurons in M1 and the ECN at each age. M1 data are the numerical values for *Figure 3—figure supplement 2*.

DOI: https://doi.org/10.7554/eLife.41841.010

and Gaussian-exponential functions (*Figure 3a,b*). Representative fits of P9 and P12 ECN neurons are shown in *Figure 6—figure supplement 1c*. In a subset of these ECN recordings, we simultaneously recorded from M1, permitting direct comparison of the responses of neurons in both structures to the same twitches and wake movements (*Figure 6—figure supplement 1d,e*). Further, at P12, we again analyzed the data using the shift predictor and found that ECN activity reliably preceded M1 activity by 15 ms (*Figure 6—figure supplement 1f*), consistent with ECN inputs projecting to the forelimb representation of M1.

Whereas ECN neurons at P9 were predominantly responsive to twitches (*Figure 7a*, left; 11 of 16 neurons), ECN neurons at P12 were responsive to both twitches and wake movements (*Figure 7a*, right; 15 of 20 neurons). Consequently, although P9 and P12 ECN neurons were similarly twitch-responsive (*Figure 7b*, blue bars; $t_{34}$ = 1.8, p=0.09), P12 neurons were significantly more wake-responsive than P9 neurons (*Figure 7b*, red bars; $t_{34}$ = 2.9, p=0.006), resulting in very different distributions at these ages (*Figure 7c*). Based on these findings, we conclude that the increase in wake responsiveness of M1 neurons at P11 and P12 is due to a state-dependent change in sensory gating in the ECN.

## Twitch-related activity in M1 is locally masked at P12

At P12, neurons in the ECN continue to respond to twitch-related reafference; accordingly, the decrease in twitch-related activity in M1 neurons cannot be attributed to the ECN. One possibility is that twitch-related reafference continues to reach M1 at P12, but changes in local circuitry—particularly local inhibitory circuitry—serve to mask this activity. If true, local disinhibition of M1 should unmask responses to twitches. Thus, we recorded M1 activity before and after local pharmacological disinhibition with the GABA$_A$ antagonist, bicuculline (n = 6 pups, 99 neurons) or injection of saline (n = 6 pups, 107 neurons; *Table 1*). We confirmed that electrodes were located within the forelimb representation of M1. Also, by adding Fluoro-Gold to the solution and using its spread as a proxy

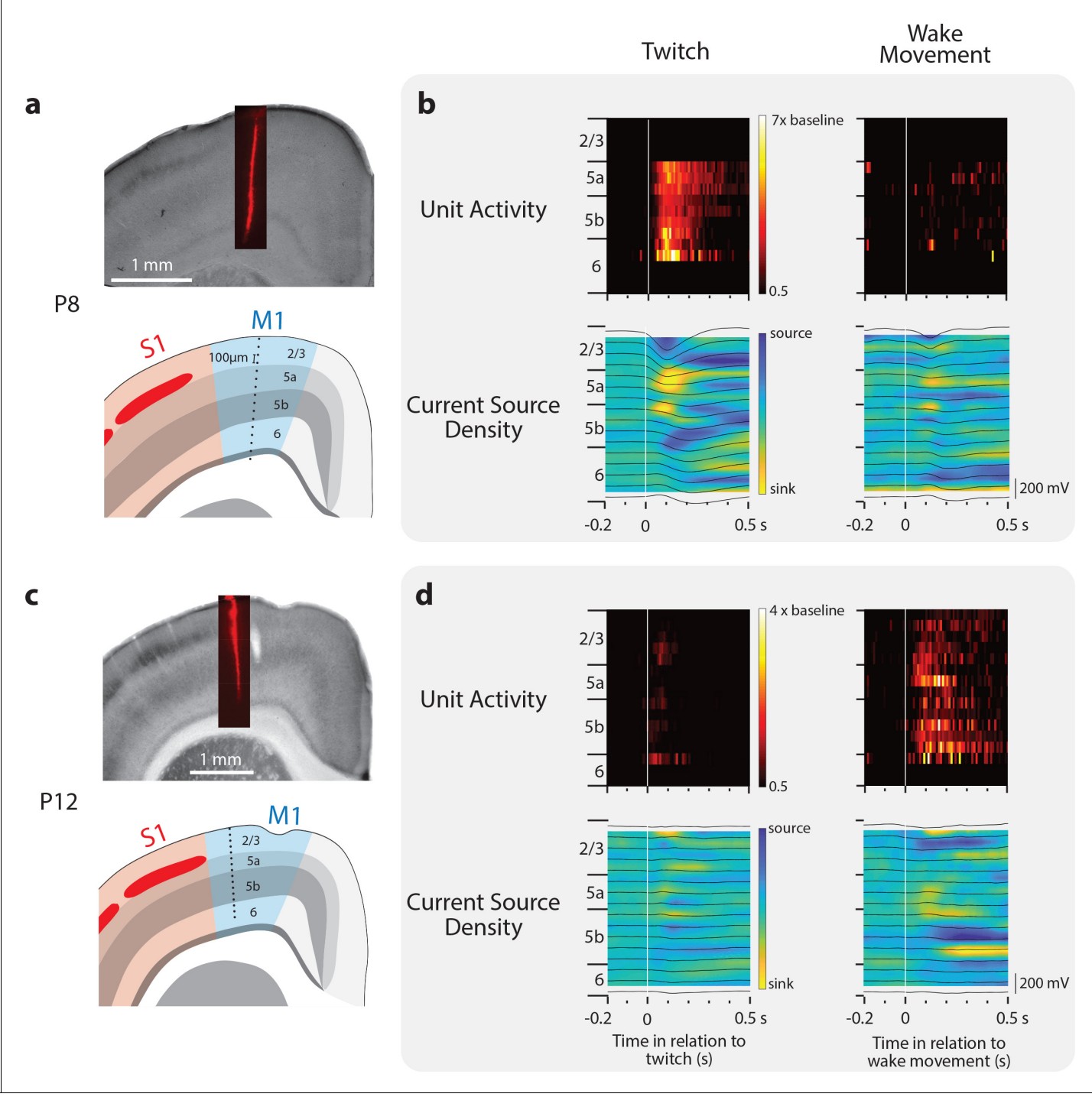

**Figure 4.** Layer-specific reafferent activity in M1. (**a**) Top: Location of laminar electrode in CO-stained sensorimotor cortex in a P8 rat. Bottom: Identification of the cortical layers of M1 along with the location of the 16 recording sites (black dots) along the electrode (site separation = 100 μm). (**b**) Top row: Neural responses (relative to baseline firing rate) for all neurons isolated at each electrode site triggered on twitches (left) and wake movements (right). Bottom: Current source density plots triggered on twitches (left) and wake movements (right). Local field potentials are superimposed (black lines). (**c**) Same as in (**a**) except at P12. (**d**) Same as in (**b**) except at P12.
DOI: https://doi.org/10.7554/eLife.41841.011

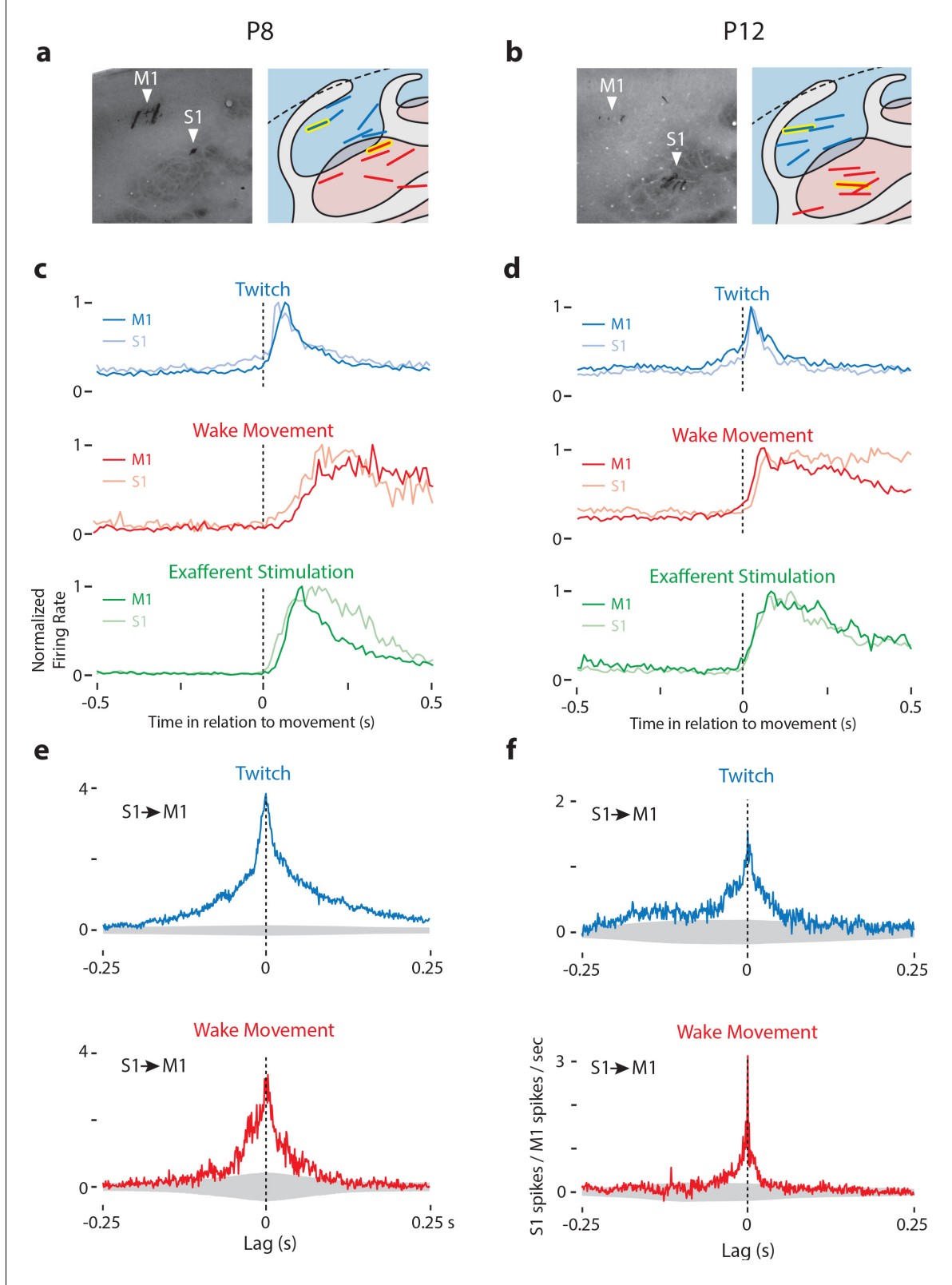

**Figure 5.** Dual recordings in M1 and S1. (**a**) Left: CO-stained tissue of the M1 and S1 forelimb representation of a P8 rat showing the location of electrodes. Right: Location of M1-S1 dual recordings for all P8 rats. M1 is shown in blue, S1 in red. Recording sites from stained tissue (left panels) are designated with yellow highlights in the right panels. (**b**) Same as in (**a**) except at P12. (**c**) Mean normalized perievent histograms (bin size = 10 ms) of neural activity of responsive neurons in M1 and S1 of P8 rats triggered on twitches (blue), wake movements (red), and exafferent stimulation (green).
*Figure 5 continued on next page*

*Figure 5 continued*

Blue plots include all twitch-responsive neurons (M1: N = 50, S1: N = 31), red plots include all wake-responsive neurons (M1: N = 21, S1: N = 18), and green plots include all twitch- or wake-responsive neurons (M1: N = 55, S1: N = 38). (d) Same as in (c) except at P12. Blue plots include all twitch-responsive neurons (M1: N = 15, S1: N = 15), red plots include all wake-responsive neurons (M1: N = 60, S1: N = 56), and green plots include all twitch- or wake-responsive neurons (M1: N = 63, S1: N = 65). (e) Cross-correlation (bin size = 1 ms) of all available pairs of responsive M1-S1 neurons, minus the shift predictor, for twitches (top row, blue) and wake movements (bottom row, red). These plots significant peaks of S1 activity at lags of 0 ms in relation to M1 activity. Gray regions denote confidence bands (p=0.01). (f) Same as in (e) except at P12.

DOI: https://doi.org/10.7554/eLife.41841.012

for the spread of bicuculline, we confirmed that drug diffusion was largely confined to the forelimb representation of M1 (*Figure 8a,b*).

Perievent histograms of twitch and wake responses before (Pre) and after (Post) injection of bicuculline or saline were fit to Gaussian and Gaussian-exponential functions. Representative neural and behavioral data during active sleep and wake, as well as representative perievent histograms, are illustrated in *Figure 8—figure supplement 1*. All bicuculline- and saline-injected pups continued to cycle between sleep and wake, and neither group showed a significant difference in the number of twitches ($F_{(1,10)} = 0.12$, p=0.74) or wake movements ($F_{(1,10)} = 4.0$, p=0.07; *Table 1*). For saline-injected pups, scatterplots of $r^2_{twitch}$ and $r^2_{wake}$ replicate our previous finding: P12 M1 neurons were predominantly wake-responsive. Likewise, before the injection of bicuculline, M1 neurons were predominantly wake-responsive (*Figure 8c*, bottom left). However, after injection of bicuculline, there was a clear increase in twitch-responsiveness of the previously wake-responsive neurons (*Figure 8c*, bottom right). This twitch-responsiveness was not the result of non-specific or abnormal bursting (*Figure 8—figure supplement 1a*). Quantitatively, as shown in *Figure 8d*, M1 neurons exhibited increased $r^2_{twitch}$ values after injection of bicuculline ($Z_{65} = 6.8$, p<0.0001), with no corresponding change in $r^2_{wake}$ ($Z_{65} = 1.8$, p=0.08); saline injection had no effect on either $r^2_{twitch}$ or $r^2_{wake}$. Because local disinhibition of M1 neurons unmasked twitch-related reafference (*Figure 8e*), it is clear that twitch-related reafference reaches M1 at P12, just as it does at P8.

## Discussion

We have documented a complex and rapid developmental transition in M1 sensory responses that occurs toward the end of the second postnatal week in rats. Consistent with previous reports, we found that M1 neurons at P8 were overwhelmingly responsive to twitch-related reafference; in contrast, reafference from wake movements failed to trigger M1 activity due to sensory gating at the ECN (*Tiriac and Blumberg, 2016*). This pattern persisted through P10. Then, at P11, M1 neurons suddenly exhibited robust responses to wake-related reafference, a developmental 'awakening' that, as demonstrated here, results from an upstream change in state-dependent sensory processing of self-generated movements in the ECN. By P12, twitch-related reafference continues to reach M1 neurons, but is less likely to drive neural activity. Further, we demonstrated that these M1 sensory responses originate in the deep layers of M1, and occur nearly simultaneously with sensory responses in S1, suggesting parallel thalamic inputs. All together, these findings establish that activity in infant M1 more closely resembles a sensory area than a motor area and reveal a complex sequence of developmental transitions in M1's reafferent responses (*Figure 9*). We propose that the activity driven by M1's early-developing sensory framework provides a foundation for its later-emerging role in motor control and learning-related plasticity.

### Developmental changes in reafferent responses to twitches

Similar to spontaneous retinal waves for the developing visual system (*Ackman et al., 2012*; *Hanganu et al., 2006*) and spontaneous cochlear activity for the developing auditory system (*Tritsch et al., 2007*), twitches provide a robust source of proprioceptive input to M1 neurons for early sensorimotor development (*Blumberg, 2015b*). In this regard, there are several features of twitches that make them more suitable than wake movements for directing activity-dependent development of the sensorimotor system. First, unlike wake movements, twitches are low-amplitude, discrete events that occur against a background of low muscle tone; these features enable individual twitches to provide high-fidelity reafferent signals to the developing brain. Second, twitches occur

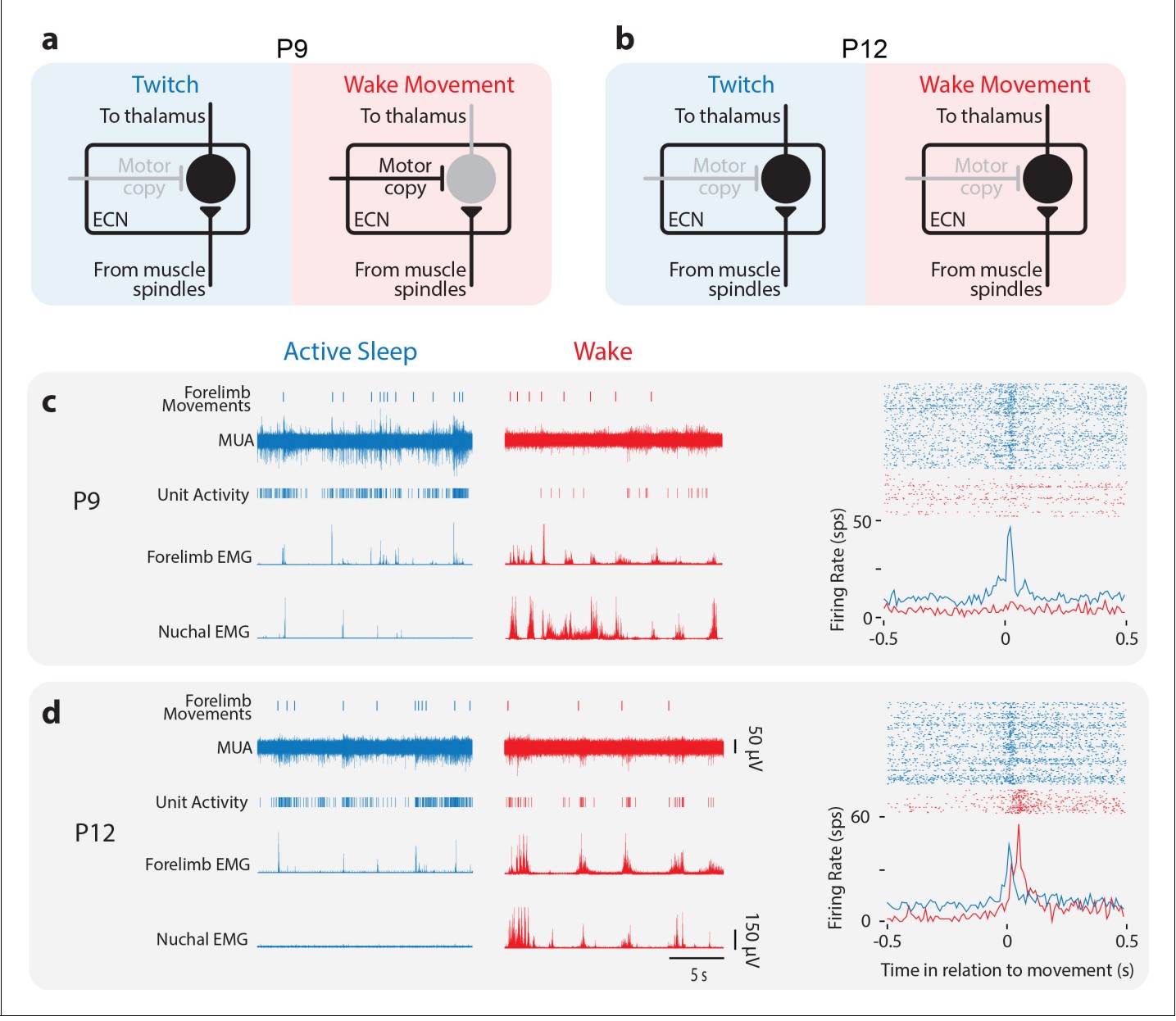

**Figure 6.** Developmental change in state-dependent ECN activity. (**a**) Model of ECN neuronal activity in response to twitches and wake movements at P9, as proposed previously (*Tiriac and Blumberg, 2016*). Neurons in the ECN convey twitch-related reafference to downstream structures, including thalamus and, ultimately, M1. For wake movements, a motor copy inhibits the ECN neuron, preventing the conveyance of reafference to downstream structures. (**b**) Proposed model of ECN neuronal activity in response to twitches and wake movements at P12. The ECN's gating of twitch-related reafference is identical to that at P9. However, at P12, we propose that wake-related reafference ceases to be gated in the ECN, permitting this reafference to be conveyed to downstream structures. (**c**) Left: At P9, representative data depicting 20 s periods of active sleep (blue) and wake (red), showing forelimb movements, multi-unit activity (MUA), sorted unit activity from the forelimb representation of M1, and rectified EMGs from ipsilateral forelimb and nuchal muscles. Right, top: Raster sweeps for an individual ECN neuron triggered on twitches (blue) and wake movements (red), with each row showing the unit activity surrounding a single movement. Right, bottom: Perievent histogram (bin size = 10 ms) showing mean firing rate for this neuron triggered on twitches (blue) and wake movements (red). (**d**) Same is in (**c**) except for a P12 rat.

DOI: https://doi.org/10.7554/eLife.41841.013

The following figure supplement is available for figure 6:

**Figure supplement 1.** ECN recording locations and representative neural activity.

DOI: https://doi.org/10.7554/eLife.41841.014

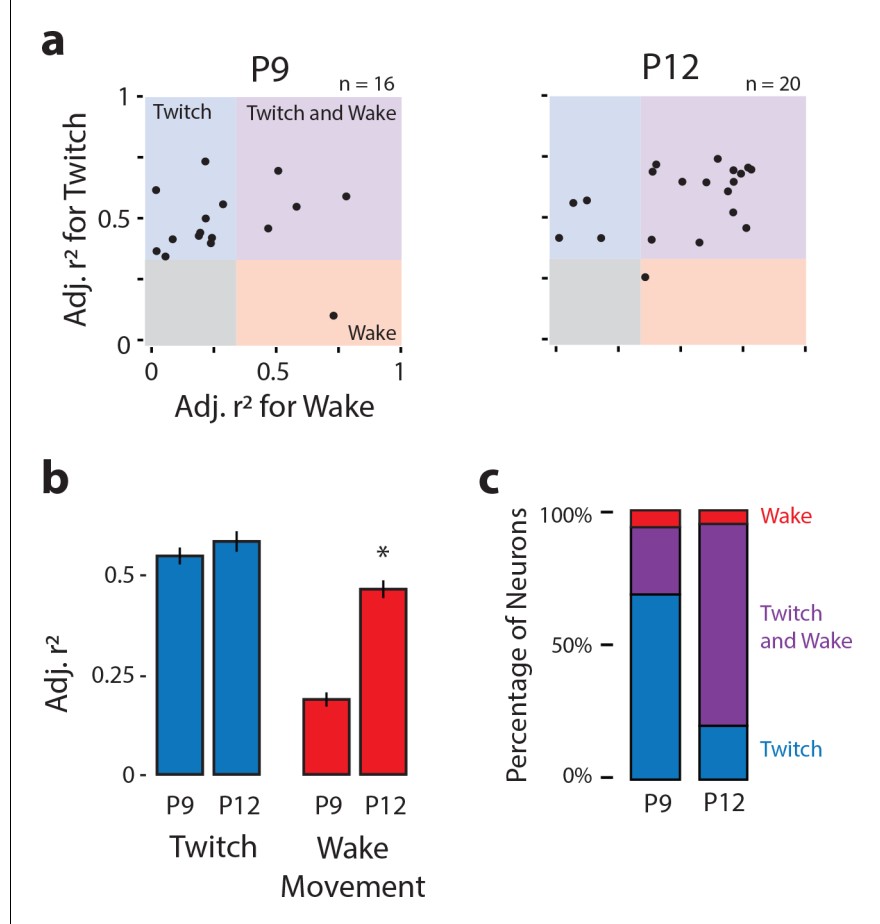

**Figure 7.** Developmental onset of wake-related reafference in the ECN. (**a**) Scatterplot of each ECN neuron's adjusted $r^2$ value for twitches (y-axis) and wake movements (x-axis) at P9 (left) and P12 (right). Background color illustrates the classification of neurons within that region of the scatterplot. (**b**) Mean (±SEM) adjusted $r^2$ values for twitches (blue) and wake movements (red) for ECN neurons recorded at P9 and P12. * Significant difference from P9 (p<0.025). (**c**) Percentage of ECN neurons at P9 and P12 that were wake-responsive (red), twitch- and wake-responsive (purple), and twitch-responsive (blue).
DOI: https://doi.org/10.7554/eLife.41841.015

exclusively and abundantly during active sleep, the predominant behavioral state in early development (*Jouvet-Mounier et al., 1970*; *Kayser and Biron, 2016*). Finally, when combined with the inhibition of wake-related reafference in ECN neurons (*Tiriac and Blumberg, 2016*), which prevents this reafference from being conveyed to neurons in M1, sensory feedback from twitches is the only reliable source of self-generated proprioceptive input to M1 neurons before P10.

Twitches are self-generated movements that typically occur in rapid succession in nearby muscles (*Blumberg et al., 2013*). Based on what is known about the receptive fields of M1 neurons in adults (*Jacobs and Donoghue, 1991*), we expected twitches from numerous forelimb muscles—not just the bicep from which we recorded—to trigger activity in M1 neurons. Because bicep twitches are preceded and followed by twitches of other forelimb muscles, and because we triggered M1 activity on each twitch event, it is difficult to disentangle the neural activity due to a bicep twitch from the activity attributable to other nearby muscles. We see evidence of this here, with neurons increasing their firing rate as early as 100 ms before the triggering twitch (e.g. see *Figure 5d*). In contrast, M1 neurons did not show similar increases in activity before wake movements, which we attribute to the fact that we triggered M1 activity only at the onset of a bout of wake movements (i.e. after a period of behavioral quiescence).

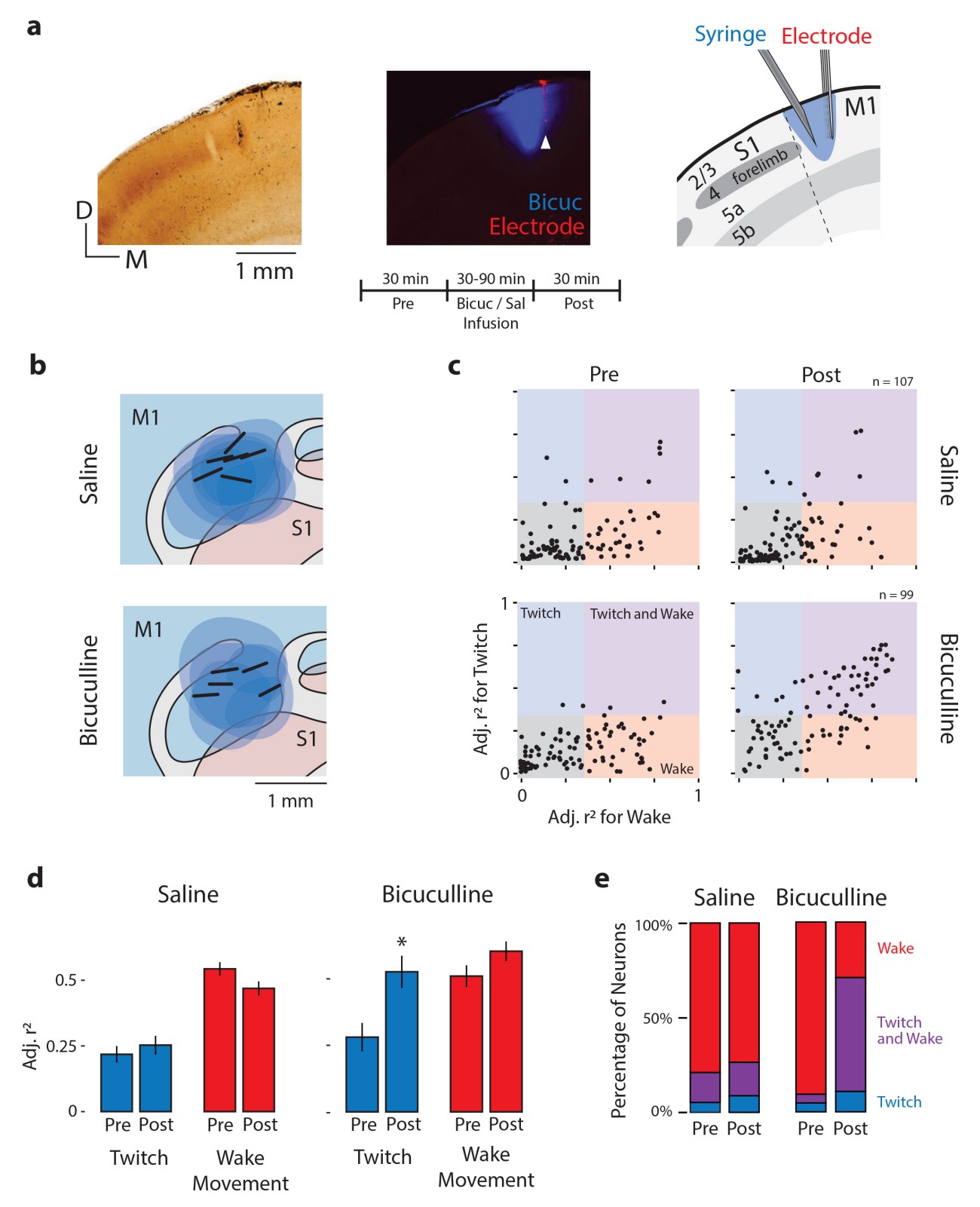

**Figure 8.** Local disinhibition of M1 neurons at P12 unmasks twitch-related activity. (**a**) Left: Coronal section showing CO staining around the S1/M1 boundary. Middle: Fluorescent image of the same section showing the spatial extent of bicuculline diffusion (blue). White arrow indicates the location of an electrode shank in M1 (red). Right: Illustration of the histological sections at left to show the boundaries of S1 and M1, the laminar structure of each area, and a reconstruction of the location of the microsyringe needle immediately lateral to the recording electrode. Middle, bottom:

*Figure 8 continued on next page*

Figure 8 continued
Experimental timeline. (b) Recording sites (black bars) and diffusion boundaries (blue ovals) for all six saline injections (top) and five bicuculline injections (bottom) depicted in the horizontal plane. The 6th bicuculine injection is shown in the coronal section in (a). All recording sites were within the forelimb representation of M1; diffusion boundaries were largely restricted to M1. (c) Top: Adjusted $r^2$ values for twitches (y-axis) and wake movements (x-axis) for each isolated M1 unit before (Pre) and after (Post) injection of saline. Bottom: Same as above, but for bicuculline group. (d) Mean (±SEM) adjusted $r^2$ values for twitches (blue) and wake movements (red) before (Pre) and after (Post) injections of saline or bicuculline. * significant difference from Pre (p<0.0125). (e) Percentage of M1 neurons that were wake-responsive (red), twitch- and wake-responsive (purple), and twitch-responsive (blue) during the Pre and Post periods in the saline and bicuculline groups.
DOI: https://doi.org/10.7554/eLife.41841.016

The following figure supplement is available for figure 8:

**Figure supplement 1.** Representative M1 neural activity before and after injection of saline or bicuculline at P12.
DOI: https://doi.org/10.7554/eLife.41841.017

Although we did not directly measure the receptive fields of M1 neurons, we are able to use the population of neurons responsive to bicep twitches to make inferences about receptive fields in M1. At P8-10, all responsive neurons were twitch-responsive and the twitches of just one muscle—the bicep—triggered responses in half of the M1 neurons from which we recorded (*Figure 3—figure supplement 1b*); this suggests that each M1 neuron's receptive field encompassed multiple forelimb muscles. Accordingly, at P8-10, we would expect twitches recorded from another forelimb muscle to yield responses in an overlapping subset of M1 neurons. Also, it should be noted that those neurons that were unresponsive to twitches and wake movements were nonetheless responsive to exafferent forelimb stimulation (*Figure 3—figure supplement 1a*), almost certainly because the method of exafferent stimulation entails stimulation of receptors throughout the limb.

The percentage of twitch-responsive neurons in M1 decreased suddenly by P12 (*Figure 3b*). At P11, of the responsive M1 neurons, 73% were twitch-responsive; by P12, that percentage had decreased to just 24% (*Figure 3c*). This sudden decrease in twitch responsiveness is attributable to local inhibition, as disinhibiting M1 using the GABA$_A$ antagonist, bicuculline, restored twitch responsiveness to 71% (*Figure 8c–e*). That M1 neurons remained twitch-responsive just 1 day earlier suggests a rapid change in the excitatory-inhibitory balance in M1 between P11 and P12, which could be due to changes in interneuron network connectivity as well as changes in the postsynaptic effects of GABA (*Ben-Ari et al., 2007*; *Blaesse et al., 2009*; *Payne et al., 2003*). Thus, by P12 some sensory inputs to M1 neurons persist beneath a layer of inhibitory control. Similar subthreshold inputs in M1 have been demonstrated previously in adults, and it has been suggested that they constitute the latent connections that permit rapid M1 plasticity (*Huntley, 1997a*; *Jacobs and Donoghue, 1991*; *Peters et al., 2017*). Accordingly, we propose that the broad receptive fields of M1 neurons at P8-10 reveal the foundation of M1's sensory framework and the maximal extent of rapid adult plasticity in that structure.

Closer inspection of twitch responses at P8 and P12 suggests a sharpening of the M1 receptive fields with age, such that a smaller subset of neurons become more consistently driven by the movement of individual muscles (*Figure 3—figure supplement 2*). In the context of inhibition, this type of sharpening—such that neurons are responsive only to preferred stimuli—has been termed the 'iceberg effect' (*Carandini and Ferster, 2000*; *Isaacson and Scanziani, 2011*; *Rose and Blakemore, 1974*) and has been demonstrated across several sensory modalities (*Liu et al., 2011*; *Poo and Isaacson, 2009*; *Wu et al., 2008*). Future work should address the specific contributions of different types of inhibitory interneurons to this developmental sharpening.

## Developmental changes in reafferent responses to wake movements

The robust sensory response of M1 neurons to twitches at P8-10 stands in sharp contrast to their conspicuous silence following wake movements (*Figure 1e*, see *Tiriac et al., 2014*)—a silence highlighted further by the sudden influx of wake-movement reafference at P11 (*Figure 1e,f*; *Figure 3d*). This transition—the result of a rapid change in the processing of wake-related reafference in the ECN—represents a milestone in the development of the sensorimotor system. Just as neurons in the developing visual cortex are initially blind to patterned light (*Colonnese et al., 2010*)—instead relying on retinal waves for early retinotopic development—neurons in M1 are initially 'blind' to the consequences of wake movements, relying instead on sensory feedback provided

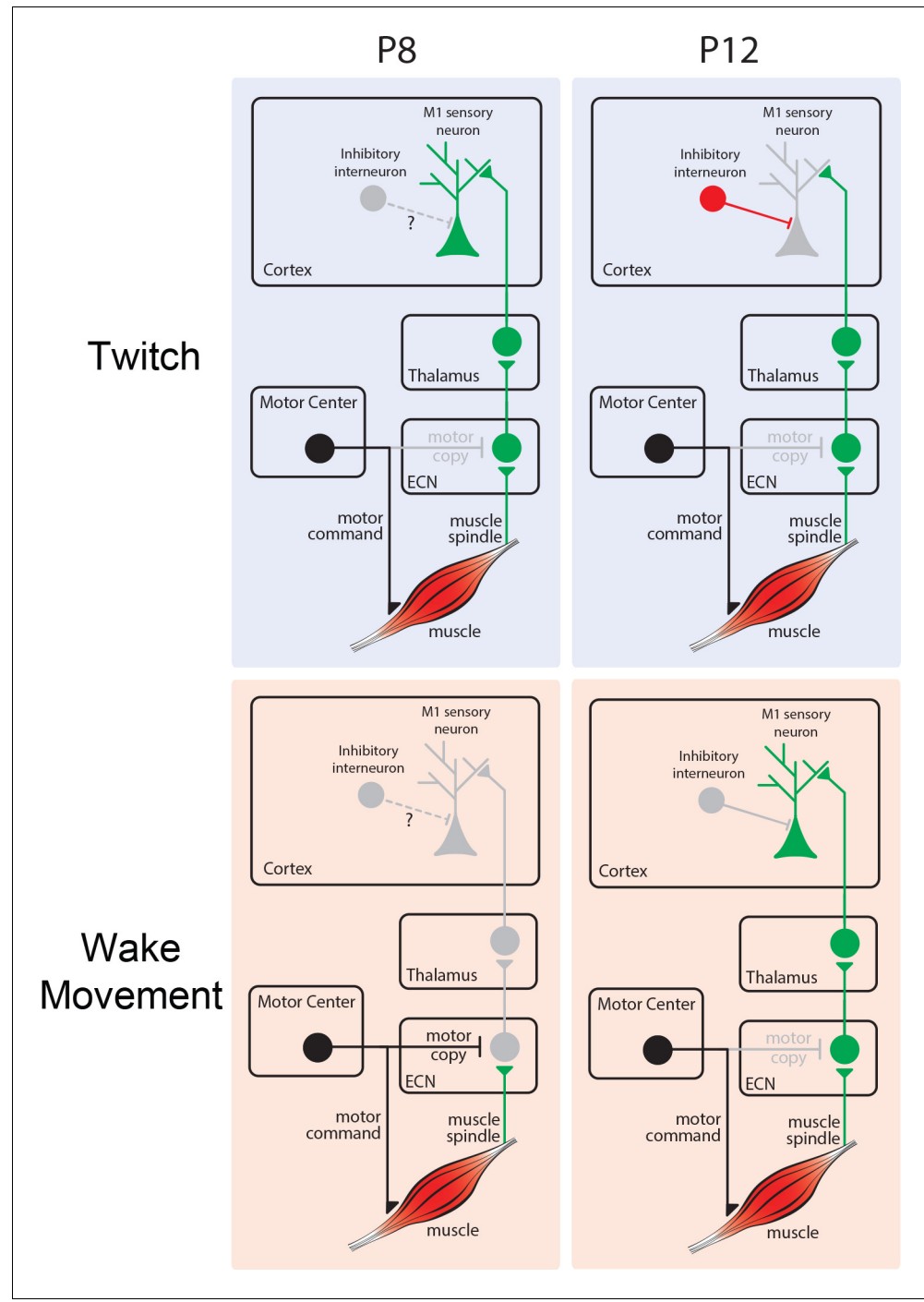

**Figure 9.** Developmental changes in reafferent processing. Top left: At P8, twitch-related reafference (green) is conveyed from ECN to downstream structures, including M1. Top right: At P12, twitch-related reafference reaches cortex, but activity is masked due to local inhibition. Bottom left: At P8, inhibition at the ECN masks wake-related reafference (green) and prevents its conveyance to M1. Bottom right: At P12, the ECN no longer selectively inhibits reafference from wake movements, permitting its conveyance to M1.
DOI: https://doi.org/10.7554/eLife.41841.018

by twitches. Interestingly, although retinal waves drive cortical activity across periods of sleep and wake (*Ackman et al., 2012*), analogous neural activity in the developing sensorimotor system occurs exclusively during active sleep.

What is the functional significance of the fact that the ECN does not relay wake movement reafference to M1 neurons before P11? Here again, we can draw upon work in the developing visual system showing that even the weakest visual input can generate a saturating response in cortex (*Colonnese et al., 2010*; for review see *Colonnese and Phillips, 2018*). If similar cortical circuitry in the developing sensorimotor system permitted twitches to generate a saturating response in M1 and S1, wake movements—if conveyed to M1—would likely result in sustained, global activation. As wake movements are not temporally restricted to individual body parts, reafference from wake movements could weaken somatotopic boundaries across sensorimotor cortex during this sensitive period of development.

Regardless, for M1 neurons to contribute to motor learning, they must eventually be sensitive to feedback from wake movements. As demonstrated here, the critical transition in the responsiveness of both M1 and ECN neurons to wake movements occurs suddenly between P10 and P12. This transition enables wake movements to differentially activate populations of M1 neurons. These early differential patterns of activation are precursors to the activity of 'movement-related neurons' in adult M1, as described by others (*Hyland, 1998*; *Komiyama et al., 2010*; *Peters et al., 2014*). Thus, the increase in wake responsiveness of M1 neurons at P12 provides an essential bridge linking their sensory activity during the first postnatal week (*Tiriac and Blumberg, 2016*; *Tiriac et al., 2014*) with their sensory activity after M1 has established its motor functions.

## Implications of wake-movement reafference for emerging motor control

In adults, the functional importance of wake-movement reafference for M1 motor control is well established (*Iriki et al., 1989*; *Iriki et al., 1991*). However, at P12, M1 neurons are not yet involved in motor control. This motivates the question: What is the function of wake-movement reafference during at this age? We propose that early wake-movement reafference shapes emerging motor control and plasticity.

Although adult M1 neurons receive a diversity of cortical and thalamic inputs, plasticity is driven through the coincident activation of both horizontal projections from S1 and ascending projections from thalamus (*Iriki et al., 1989*; *Iriki et al., 1991*). At P12, horizontal projections to M1 neurons are present, but the synaptic connections from superficial to deeper layers are silent; because of these 'silent synapses,' S1 input to M1 is incapable of driving corticospinal neurons (*Anastasiades and Butt, 2012*). However, as demonstrated here, wake-movement reafference drives activity in deep M1 neurons via ascending (i.e. thalamic) projections. Accordingly, the ascending source of reafference described here may contribute to the functional conversion of the previously silent synapses.

Once both horizontal and ascending reafferent pathways to the deep layers of M1 are active, the next stage of M1 development can begin. Intracortical microstimulation experiments in rats reveal that the corticospinal tract is sufficiently developed to produce M1-mediated movements as early as P13, although inhibition on corticospinal neurons appears to limit motor outflow for several additional weeks (*Young et al., 2012*). Thus, for motor outflow to occur, inhibition of M1 corticospinal neurons must be overcome by sufficiently strong excitation. We suggest that repeated reafference arising from subcortically generated wake movements provides synapses on deep M1 neurons the opportunity to be shaped and strengthened, producing increased excitatory drive onto corticospinal neurons. With sufficient repetition, this excitatory drive can allow M1 motor neurons to contribute to movement.

## Evolutionary implications

The traditional designation of primary motor and sensory cortices as exclusively 'motor' and 'sensory,' respectively, is contradicted by neurophysiological and comparative studies (*Asanuma, 1981*; *Baldwin et al., 2017*; *Baldwin et al., 2018*; *Cooke et al., 2015*; *Hatsopoulos and Suminski, 2011*; *Lende, 1963a*; *Matyas et al., 2010*). Further, such simplifications ignore the realities of thalamocortical loops as well as the ubiquity of descending motor projections from parietal and frontal cortex (*Nudo et al., 1995*; *Sherman, 2016*). From a comparative perspective, placental mammals have an

M1, but many marsupials do not (*Kaas, 2004*). For example, Virginia opossums have what has been referred to as a 'sensorimotor amalgam' that exhibits a mix of features of S1 and M1 as defined in placental mammals (*Karlen and Krubitzer, 2007*; *Lende, 1963a*; *Lende, 1963b*; *Lende, 1963c*). This has led to the hypothesis that S1 and M1 are derived from the same ancestral brain area and that opossums have retained this ancestral state (*Beck et al., 1996*).

Even though M1 in rats is highly specialized, its shared evolutionary history with somatosensory cortex suggests that it should develop similarly to other somatosensory areas, including S1. This suggestion is consistent with a developmental-evolutionary perspective, according to which evolution enables phenotypic transformations in cortical structure through alterations in developmental processes (*Krubitzer and Dooley, 2013*). Thus, earlier in development, M1 and S1 should exhibit more shared features. This idea gains support here from the similar sensory response profiles observed in M1 and S1 and the evidence that both sensory input to both structures arises from parallel ascending pathways.

## Conclusions

The present findings strongly support the idea that M1 is built on a sensory framework that scaffolds its later-emerging motor map (*Chakrabarty and Martin, 2005*; *Huntley, 1997b*; *Keller et al., 1996*). Unique to the present investigation, we have identified the relative contributions of sensory feedback from self-generated sleep and wake movements to M1's somatotopically organized activity. Unlike the developing visual system, behavioral state provides a critical context for characterizing the sensory development of M1 neurons. Further, our findings suggest that M1's sensory framework lays a foundation for this plasticity in adulthood.

Damage to M1 in adults can cause a profound loss of motor function. After stroke, the current therapeutic approach is to focus primarily on restoring motor control. Alternatively, it has been suggested that therapeutic outcomes for stroke patients would improve if closer attention were paid to also assessing and restoring proprioceptive function (*Semrau et al., 2015*). This suggestion aligns nicely with the present findings that M1's motor functions rest atop an earlier-developing sensory framework. Moreover, if understanding the mechanisms of normal development can help inform therapies that promote recovery following stroke (*Johnston, 2009*; *Murphy and Corbett, 2009*), then the present results suggest that the degree of early sensory recovery will predict eventual motor recovery and inform therapeutic interventions.

Finally, it should be noted that the early, intimate connection between active sleep and M1 sensory activity continues into adulthood as a connection between active sleep and motor plasticity. For example, in juvenile and adult mice, active sleep appears to critically influence the elimination and stabilization of new dendritic spines in M1 formed in a motor learning task (*Li et al., 2017*). This function is not restricted to mammals. For example, in finches, consolidation of song motor memories also depends on neural processes during active sleep (*Brawn et al., 2010*; *Derégnaucourt et al., 2005*). Such findings fit within a broader context linking active sleep to developmental plasticity and memory consolidation (*Blumberg and Dooley, 2017*; *Diekelmann and Born, 2010*; *Dumoulin Bridi et al., 2015*; *Maquet et al., 2000*). Thus, all together, the present findings encourage a new conceptualization of how M1 is functionally organized and how it adapts to and supports learning across the lifespan.

## Materials and methods

### Experimental models

For recordings in the forelimb representation of M1, a total of 49 male and female Sprague-Dawley Norway rats (*Rattus norvegicus*) at P8-12 were used (n = 7–15 at each age; mean = 15.4 neurons per pup, s.d. = 7.24). In a subset of these animals, recordings were also performed in the ECN (n = 6 at P8-9; n = 7 at P11-12) or the forelimb representation of primary somatosensory cortex (S1; n = 6 at P8 and P12). Additional laminar electrode recordings were performed at P8 and P12 (n = 2 at each age). For M1 recordings before and after injection of saline or bicuculline, a total of 12 male and female rats were used at P12 (n = 6 per group). See *Table 1* for additional information.

Pups were born to mothers housed in standard laboratory cages (48 × 20 × 26 cm) in a room with a 12:12 light dark schedule. Food and water were available ad libitum. Expecting mothers were

checked at least once daily for pups. The day of birth was considered P0. If necessary, on or before P3 litters were culled to eight pups (typically with equal numbers of males and females). Littermates were never assigned to the same experimental groups. All experiments were conducted in accordance with the National Institutes of Health (NIH) Guide for the Care and Use of Laboratory Animals (NIH Publication No. 80–23) and were approved by the Institutional Animal Care and Use Committee of the University of Iowa.

## Surgery

For all studies, pups were prepared for neurophysiological recording using methods similar to those described previously (*Blumberg et al., 2015a*; *Tiriac and Blumberg, 2016*; *Tiriac et al., 2014*). On the day of testing, a pup with a visible milk band was removed from the litter. Under isoflurane anesthesia (3–5%; Phoenix Pharmaceuticals, Burlingame, CA), custom-made bipolar hook electrodes (epoxy coated, 0.002 inch diameter; California Fine Wire, Grover Beach, CA) were implanted into the *biceps brachii* muscle of the forelimb, the *extensor digitorum longus* muscle of the hindlimb, and the nuchal muscle for electromyographic (EMG) recordings. Wires were secured using a small amount of collodion. A stainless-steel ground wire (uncoated, 0.002 inch diameter; California Fine Wire, Grover Beach, CA) was implanted transdermally on the dorsum. The pup was injected with carprofen (5 mg/kg subcutaneously; Putney, Portland, ME) and a rectangular section of skin was removed above the skull. After topical application of bupivicane as an analgesic (Pfizer, New York, NY), the skull was cleaned and dried. Vetbond (3M, Minneapolis, MN) was applied to the skin around the perimeter of the exposed skull and a custom-built head-fix apparatus was secured to the skull using cyanoacrylate adhesive. To limit mobility during recovery, the pup was wrapped in gauze and maintained at thermoneutrality (35°C) in a humidified incubator for at least 1 hr. The entire surgery lasted approximately 15 min.

After recovery, the pup was lightly reanesthetized (2–3% isoflurane) in a stereotaxic apparatus. A small hole (diameter = 1.8 mm) was drilled in the skull using a trephine drill bit (1.8 mm; Fine Science Tools, Foster City, CA), leaving the dura intact. In experiments where saline or bicuculine were also injected into M1, a larger 2.7 mm trephine was used to allow placement of both the electrode and the microsyringe needle. For M1 forelimb recordings, the coordinates were: AP: 0.8–1.2 mm anterior to bregma; L: 1.7–2.0 mm. For S1 forelimb recordings, the coordinates were: AP: 0.2–1 mm anterior to bregma; L: 2.2–3.5 mm. For ECN recordings, the coordinates were: AP: 3.0–3.2 mm posterior to lambda; L: 1.6–2.0 mm. Small holes were also made bilaterally in occipital cortex to allow insertion of the thermocouple and combined reference/ground electrode. This procedure lasted approximately 5 min, after which anesthesia stopped and the exposed dura was covered with mineral oil to prevent drying. The pup was then transferred and secured to a different stereotaxic apparatus within a Faraday cage where its torso was supported on a narrow platform and its limbs dangled freely on both sides (*Figure 1a*). Brain temperature was monitored using a fine-wire thermocouple (Omega Engineering, Stamford, CT) inserted into occipital cortex contralateral to the M1 recording site. The pup acclimated for at least 1 hr until its brain temperature reached at least 36°C and it was cycling between sleep and wake, at which time electrophysiological recordings began.

## Electrophysiological recordings

The EMG bipolar hook electrodes and corresponding ground electrode were connected to a differential amplifier (Tucker-Davis Technologies, Alachua, FL). A chlorinated Ag/Ag-Cl ground electrode (0.25 mm diameter; Medwire, Mt. Vernon, NY) was inserted into occipital cortex ipsilateral to the M1 recording site or contralateral to the ECN recording site. For M1 and S1 recordings, data were acquired using 16-channel silicon depth electrodes (*Figure 1—figure supplement 1a* bottom; Model A4x4-3mm-100-125-177-A16; NeuroNexus, Ann Arbor, MI). The electrodes were positioned to be perpendicular to the cortical surface (0° to 20° medial, depending on location) and inserted 700–1400 µm beneath the cortical surface, corresponding to the deeper layers of cortex. For laminar recordings, a single shank 16-channel electrode (Model A1x16-3mm-100-703-A16) was used. Linear electrodes were inserted approximately 1500 µm perpendicular to the cortical surface. For ECN recordings, custom-designed 16-channel silicon depth electrodes were used (*Figure 4c*; NeuroNexus). The electrode was angled caudally 14–16° and lowered 3.7–4.2 mm beneath the surface of the brain. EMG and neural signals were sampled at approximately 1 kHz and 25 kHz, respectively.

Bandpass filters were applied to EMG (300–5000 Hz) and neural (0.1 Hz-12.5 kHz) signals. A notch filter was also used. Before insertion of a silicon electrode, it was coated with fluorescent DiI (Vybrant DiI Cell-Labeling Solution; Life Techologies, Grand Island, NY) for subsequent histological verification of placement. Depth of insertion was monitored using a hydraulic micromanipulator (FHC, Bowdoinham, ME).

## General experimental procedure

The M1 electrode was slowly lowered while manually stimulating the contralateral forelimb (or the ipsilateral forelimb for ECN recordings) using a small wooden probe until a neural response was detected. Once responsive neurons were identified, the electrode settled in place for at least 15 min to allow stabilization of neural signals before the start of data collection. Recording sessions comprised continuous collection of neurophysiological and EMG data for 30 min. During acquisition, the experimenter monitored the pup's behavior and used two digital markers to record the occurrence of active-sleep twitches and wake movements of the forelimb of interest. As described previously (*Karlsson et al., 2005*), myoclonic twitches are phasic, rapid, and independent movements of the skeletal muscles against a background of muscle atonia (*Figure 2*, *Figure 6c,d*, *Figure 8—figure supplement 1a*). In contrast, wake movements are high-amplitude, coordinated movements occurring against a background of high muscle tone (*Figure 2*, *Figure 6c,d*, *Figure 8—figure supplement 1a*). Throughout the recording session, the behavior of the animal was also recorded using a digital camera (Prosilica GC; Allied Vision, Exton, PA) whose signal was acquired and synchronized with the electrophysiological record (RV2; Tucker-Davis Technologies).

After 30 min of uninterrupted behavioral data collection, exafferent neural responses were recorded by using a wooden probe to produce an elbow flexion. This method of exafferent stimulation was meant to activate proprioceptors in the bicep muscle, which was the muscle used to detect forelimb twitches and wake movements. However, in addition to providing tactile stimulation, this method certainly activated proprioceptors in other forelimb muscles. At least 20 stimulations were presented and were spaced 5–10 s apart.

## Pharmacological disinhibition of forelimb M1

Pups were prepared for M1 recording as described above. In addition, a Hamilton microsyringe (1 μL; Hamilton, Reno, NV) was inserted immediately lateral to the recording electrode (see *Figure 8b*). The recording session began with a 30 min baseline period (Pre) followed by a 30- to 90 min diffusion/acclimation period and a 30-min recording period (Post). A total volume of 0.25–0.3 μL of bicuculline (2 mM; Sigma-Aldrich, St. Louis, MO) or saline was injected slowly (0.1 μL/min). Fluoro-Gold (4%; Fluorochrome, Denver, CO) was also included in both saline and bicuculline injections to enable subsequent visualization of drug diffusion (*Figure 8b*). Immediately after injection of bicuculline, M1 multiunit activity became synchronized, with all neurons bursting every 300–400 ms. The period of this activity varied across pups and the Post period did not begin until normal multiunit activity was observed. The bursts of synchronized activity rarely occurred after the Post period began; when they did, activity within ±100 ms of the burst was not included in the analysis.

## Histology

At the end of data collection, the pup was euthanized with an overdose of 10:1 ketamine/xylazine (>0.08 mg/kg) and perfused with phosphate-buffered saline (PBS), followed by 4% paraformaldehyde. The brain was immediately extracted and post-fixed in 4% paraformaldehyde for at least 24 hr. Next, 24–48 hr before the brain was sectioned, it was transferred to a 20% solution of sucrose in PBS until it was no longer buoyant in solution.

For all but three brains with four shank M1 and S1 recordings, the cortical hemispheres were dissected apart from the underlying tissue, including the hippocampus and basal ganglia, and flattened between glass slides separated by 1.5 mm copper-coated zinc spacers (United States Mint, Washington, D.C.) for 5–30 min. Small weights (10 g) were used to apply light pressure to the top glass slide. The flattened cortex was then sectioned tangential to the pial surface. In the remaining three pups, and in all animals where laminar recordings were performed, the cortex was sectioned coronally to confirm electrode depth (*Figure 4*, *Figure 1—figure supplement 1*). Regardless of the plane of section, cortex was sectioned at 80 μm using a freezing microtome (Leica Microsystems, Buffalo Grove,

IL). To confirm the location of medullary recordings in the ECN, the medulla was sectioned coronally at 80 µm. Electrode location and drug diffusion were initially visualized and photographed in free-floating sections at 2.5X, 5X, or 10X using a fluorescent microscope and digital camera (Leica Microsystems).

Cortical sections were stained for cytochrome oxidase (CO), which has been shown in developing rats as young as P5 to reliably delineate primary sensory areas, including S1 (*Seelke et al., 2012*). Briefly, cytochrome C (3 mg per 10 mL solution; Sigma-Aldrich), catalase (2 mg per 10 mL solution; Sigma-Aldrich) and 3,3'-diaminobenzidine tetrahydrochloride (DAB; 5 mg per 10 mL solution; Spectrum, Henderson, NV) were dissolved in a 1:1 dilution of PB-$H_2O$ and distilled water. Sections were developed in well plates on a shaker at 35–40°C for 3–6 hr after which they were washed and mounted. Medullary sections were stained with cresyl violet.

Stained sections were again photographed at 2.5X or 5X magnification, combined into a single composite image (Microsoft Image Composite Editor; Microsoft, Redmond, WA), and the location of the electrode was visualized in relation to areal, nuclear, or laminar boundaries of the stained tissue.

## Statistical analysis

All analyses and statistical tests for neural and behavioral data were performed using custom-written MATLAB routines (version 2017a; Mathworks, Natick, MA) and Spike2 software (version 8; Cambridge Electronic Design). Alpha was set at 0.05 for all analyses, unless otherwise stated. Normally distributed data were tested for significance using a one-way ANOVA, one-way repeated measures ANOVA, or t-test. Non-normally distributed data (adjusted $r^2$ values, model fit parameters) were tested for significance using the Kruskal-Wallis nonparametric test or the Mann-Whitney U test. When Kruskal-Wallis tests were significant, post hoc pairwise comparisons across age were performed using the Mann-Whitney U test, with significance values adjusted for multiple comparisons using the Bonferroni procedure. Reported group data in text are always mean ±standard error (SEM), unless otherwise stated. Box plots represent the $25^{th}$, $50^{th}$ (median), and $75^{th}$ percentiles. Datapoints were considered outliers if they differed from the median by more than three standard deviations.

## Behavioral state and movement classification

As described previously (*Tiriac et al., 2014*), EMG signals and digitally scored behavior were used to identify behavioral states. EMG signals were rectified and smoothed at 0.001 s. Periods of wake were identified by dichotomizing all available EMGs into periods of high tone (indicative of wake) and atonia (indicative of sleep). As the nuchal muscle EMG typically shows chronic tone during wake periods, it was used most often for classifying sleep and wake states. Active sleep was characterized by the occurrence of myoclonic twitches against a background of muscle atonia (*Seelke and Blumberg, 2008*). For each EMG, a twitch threshold was set that was at least 3X greater than baseline. The initiation time of twitches was recorded as the first data point where the rectified EMG signal exceeded this threshold. When a twitch was identified, a subsequent twitch of that same muscle was only counted if it occurred at least 300 ms after the first; this procedure protected against duplicative analysis of neural data.

For identification of forelimb wake movements, the forelimb EMG was rectified and smoothed at 0.01 s. The baseline wake EMG value was then calculated and a threshold of 5X this baseline value was then determined. To be considered a wake movement, the smoothed EMG waveform had to be preceded by a period of low tone, rise and remain above the wake threshold for at least 300 ms, and then either be (1) behaviorally scored as a forelimb wake movement during data collection or (2) identified as a forelimb wake movement by reviewing the video recording. The wake movement's initiation time was set as the first data point where the rectified and smoothed EMG waveform exceeded the threshold. This criterion for the initiation of wake movements was established in preliminary experiments by observing video of pup forelimb movements simultaneously with EMG records; in this way, we could identify the time at which changes in the EMG waveform most reliably predict limb movements on the video.

Our criteria ensured that only the first wake movement in a bout of wake movements was used as a trigger for assessing associated neural activity (although all wake movements were scored

behaviorally). Thus, the counts of triggered wake movements underestimated the actual number of behaviorally scored wake movements across the 30-min recording periods (*Figure 1—figure supplement 2a*).

Using experimenter-recorded digital markers, EMG signals, and video, the timing of exafferent stimulation was also identified.

## Multiunit activity and spike sorting

All electrode channels were filtered for multiunit activity (MUA; 500–5000 Hz). Spike sorting was performed offline using Spike2. Once filtered, movement artifact was still visible in some medullary recordings; to remove this artifact we created a virtual reference for each channel out of the mean waveform of the remaining channels (*Ludwig et al., 2009*). This virtual reference was then subtracted from each channel's multiunit waveform, thereby removing the movement artifact.

Spike sorting was performed on channels with visually identifiable spiking activity that exceeded at least 2X the noise band using template matching with a 1.3 ms window; templates were further refined using principal components analysis in Spike2. All putative neurons were visually investigated and waveforms were excluded as outliers when they were more than 3.5 standard deviations beyond the mean of a given template. Outliers were rare and typically the result of electrical artifact. Representative waveforms can be found in the insets for all example neurons provided throughout the text (*Figure 2c*, *Figure 6—figure supplement 1c*, *Figure 8—figure supplement 1b*).

Neurons were only included in the final analysis if their signal remained stable throughout the recording session and they could be clearly differentiated from background noise (*Figure 1—figure supplement 1c,d*). An analysis of the amplitude of all isolated cortical neurons revealed no systematic differences in the stability of neurons between P8 and P12, with the peak amplitude waveform templates appearing similar (i.e., ±5%) at the beginning and end of the recording sessions. To ensure that two channels did not identify the same putative neuron, cross-correlations of all putative neurons were graphed and investigated; duplicate neurons were excluded when necessary.

## Local field potential and current source density

Raw waveforms were smoothed ($\tau$ = 1 ms) and downsampled to 1000 Hz. These waveforms were there averaged across all twitches and wake movements in order to generate sensory evoked potentials (*Figure 4b,d* lower; black lines). For P8 rats, we restricted our analysis to periods in which twitches clearly triggered local field potential events. To calculate a current source density map, a custom MATLAB script was used that determined the second spatial derivative of sensory evoked potentials in each channel, and interpolated the resulting data (*Figure 4b,d*).

## Movement-related neural activity

The relationship between neural response type and firing rate was assessed as follows: First, using either twitches or wake movements as triggers, perievent histograms (2000 ms windows; 10 ms bins) of neural activity were constructed. The average firing rate (in spikes per second, sps) was calculated for each bin and plotted. Next, the twitch-triggered perievent histogram was fit to the following Gaussian model:

$$R(t) = BL + \left( R_{max} \cdot e^{-\frac{(t-t_{max})^2}{2c^2}} \right)$$

with baseline term $BL$ in sps, maximum response term $R_{max}$ in sps, maximum time term $t_{max}$ in seconds, and Gaussian width term $c$, proportunate to the half-width at half-height ($HWHH$), such that:

$$HWHH = \sqrt{2\ln(2)}c$$

The rising phase of the neuron's wake response was fit to the same Gaussian function above, whereas the fall was fit to an exponential decay function, centered around the time of maximal response:

$$R(t) = \begin{cases} t \leq t_{max}, & BL + \left( R_{max} \cdot e^{-\frac{(t-t_{max})^2}{2c^2}} \right) \\ t > t_{max}, & BL + \left( R_{max} \cdot e^{-\lambda \cdot (t-t_{max})} \right) \end{cases}$$

with the same terms as for the Gaussian model, but with the addition of the exponential term λ, proportional to the half-life $t_{\frac{1}{2}}$, where

$$t_{\frac{1}{2}} = \frac{\ln(2)}{\lambda}$$

The r²$_{adj}$ of these models was then calculated using the following equation:

$$r^2_{adj} = 1 - \left(\frac{n-1}{n-p}\right)\frac{SSE}{SST}$$

where $n$ is the number of observations, p is the number of parameters, $SSE$ is the sum of squared error, and $SST$ is the sum of squared total.

By selecting a threshold r²$_{adj}$ value for twitch and wake movement model fits, every neuron was classified as one of four types: unresponsive, twitch-responsive, wake-responsive, or twitch- and wake-responsive. To select this threshold, mean perievent histograms were plotted for each age and neural classification. Using an r²$_{adj}$ value of 0.35, unresponsive neurons were most effectively differentiated from responsive neurons (*Figure 3—figure supplement 1a*). We confirmed that small changes (±0.15) to this threshold value did not meaningfully change the categorization of responsive neurons. Thus, neurons were considered responsive to a given movement (twitch or wake) if they had an adjusted r²$_{adj}$ > 0.35.

For responsive neurons, these models also provided estimates of baseline firing rate, peak time, peak firing rate, half-width at half-height, and, for wake movements, half-life.

## Neural percentages

The percentage of responsive neurons ($P(R_{total})$) was calculated to be the sum of neurons with either $r^2_{twitch}$ or $r^2_{wake}$ greater than 0.35, divided by the total number of neurons ($N$), as follows:

$$P(R_{total}) = \left(\frac{\sum\left(N\left(r^2_{twitch} > 0.35\right) | N\left(r^2_{wake} > 0.35\right)\right)}{\sum N}\right) * 100$$

The percentage of twitch-responsive, wake-responsive, and twitch- and wake-responsive neurons were then calculated relative to the total number of responsive neurons.

## Percentage of movements with a response

For both twitches and wake movements, triggered movements did not always result in a neural response. Thus, we quantified the percentage of movements (with each twitch or wake movement being an individual trial) that resulted in more activity than would be expected during baseline activity. First, we used the models described above to determine the expected response start time ($t_{start}$) and end time ($t_{end}$) relative to the movement time for each neuron. Next, we computed the response duration ($t_{response}$), such that:

$$t_{response} = t_{end} - t_{start}$$

To calculate the total response for each movement ($R_{trial}$), we then summed the total number of action potentials ($S$) within this response window for each movement:

$$R_{trial} = \left(\sum_{t_{end}}^{t_{start}} S\right)$$

Next, we determined the proportion of movement trials ($P(R_{trial\ raw})$) with more action potentials than what would be expected at the baseline firing rate:

$$P(R_{trial\ raw}) = \frac{\sum\left(R_{trial} > \left(BL * t_{response}\right)\right)}{N_{movements}}$$

Because the baseline action potentials were not normally distributed, we then calculated the number of action potentials during a period of baseline activity starting 2 s before the movement ($R_{BL}$):

$$R_{BL} = \left( \sum_{t=\left(-2+t_{response}\right)}^{t=-2} S \right)$$

and determined the proportion of baseline periods of duration $t_{response}$ that exceeded the expected baseline firing rate:

$$P(R_{BL}) = \frac{\sum \left( R_{BL} > \left( BL * t_{response} \right) \right)}{N_{movements}}$$

Finally, we subtracted the proportion of trials where the baseline firing rate exceeded its expected value ($P(R_{BL})$) from the proportion of trials with a response above baseline ($P(R_{trial\,raw})$), and multiplied this value by 100 to determine the percentage of trials with a response exceeding the baseline firing rate ($P(R_{trial})$):

$$P(R_{trial}) = \left( \frac{P(R_{trial\,raw}) - P(R_{BL})}{1 - P(R_{BL})} \right) * 100$$

## Cross-correlations

For M1–S1 and M1–ECN dual recordings, we calculated cross-correlations for all available pairs of responsive neurons. These cross-correlations were computed using the activity of the first neuron (in S1 or ECN) triggered on the activity of an M1 neuron within 200 ms of either a twitch or wake movement. The resulting raw cross-correlation contained activity from two potential sources: (1) correlated activity due to the stimulus (twitch or wake movement) and (2) correlated activity due to neuron-neuron interactions (i.e., between S1/ECN and M1). We used the shift predictor to remove correlations due to the stimulus and reveal correlations due to neuron-neuron interactions (*Averbeck and Lee, 2004*; *Engel et al., 1990*; *Perkel et al., 1967*). Specifically, the shift predictor is defined as the mean cross-correlation produced by all noncongruent stimulus presentations (i.e. the cross-correlation is computed for Neuron A's activity in relation to Twitch Z and Neuron B's activity in relation to all twitches except Z). For bin τ, the shift-predictor-corrected cross-correlation for neurons 1 and 2 to stimulus $s$ is defined as:

$$CC_{12s}(\tau) = \langle r_{1s}(t+\tau)r_{2s}(t) - m_{1s}(t+\tau)m_{2s}(t) \rangle_t$$

with response $r$ and mean response $m$ across time bins $t$. The resulting shift-predictor-corrected cross-correlation was summed for all available neuron pairs ($t = \pm250$ ms in 1-ms bins).

Confidence bands (p=0.01) for the resulting cross-correlogram were calculated by assuming that the count for each bin of the shift predictor follows a Poisson distribution (in which the standard deviation equals the square root of the total). Using the standard deviation, we then determined whether the difference between the raw cross-correlation and the shift predictor for that bin differed more than would be expected by chance. Values outside the confidence band indicate a source of cross-correlation beyond that induced by the stimulus and are interpreted as evidence of neuron-neuron interactions.

## Data and software availability

Whenever possible, individual data points are represented within a figure and summarized in the included tables. Raw data time series for each animal (neural firing timecodes and behavioral event timecodes) have been uploaded to Dryad (https://dx.doi.org/10.5061/dryad.8231nj1).

Custom MATLAB scripts for generating and fitting perievent histograms to twitch and wake movement models can be found on github (https://github.com/jcdooley/Dooley_and_Blumberg_2018; copy archived at https://github.com/elifesciences-publications/Dooley_and_Blumberg_2018).

## Acknowledgements

We thank Leah Krubitzer, Dylan Cooke, Greta Sokoloff, and Michaela Donaldson for helpful comments. Roustem Khazipov and Azat Nasretdinov generously provided their MATLAB script for

calculating current source density. This research was supported by grants from the National Institutes of Health (R37-HD081168 to MSB and F32-NS101858 to JCD).

## Additional information

### Funding

| Funder | Grant reference number | Author |
|---|---|---|
| National Institutes of Health | R37-HD081168 | Mark S Blumberg |
| National Institutes of Health | F32-NS101858 | James C Dooley |

The funders had no role in study design, data collection and interpretation, or the decision to submit the work for publication.

### Author contributions

James C Dooley, Conceptualization, Data curation, Software, Formal analysis, Funding acquisition, Validation, Investigation, Visualization, Methodology, Writing—original draft, Project administration, Writing—review and editing; Mark S Blumberg, Conceptualization, Resources, Supervision, Funding acquisition, Investigation, Visualization, Methodology, Writing—original draft, Project administration, Writing—review and editing

### Author ORCIDs

James C Dooley http://orcid.org/0000-0002-9868-9840
Mark S Blumberg http://orcid.org/0000-0001-6969-2955

### Ethics

Animal experimentation: All experiments were conducted in accordance with the National Institutes of Health (NIH) Guide for the Care and Use of Laboratory Animals (NIH Publication No. 80-23) and were approved by the Institutional Animal Care and Use Committee of the University of Iowa (protocol # 7011955).

### Decision letter and Author response

Decision letter https://doi.org/10.7554/eLife.41841.023
Author response https://doi.org/10.7554/eLife.41841.024

## Additional files

### Supplementary files

• Transparent reporting form
DOI: https://doi.org/10.7554/eLife.41841.019

### Data availability

Data represented in all figures is summarized in the included tables. Because of the large amount of data in the present publication (over 1,000 neurons across over 50 animals, along with thousands of behaviorally scored twitches and wake movements) our raw data (neural firing timecodes and behavioral event timecodes) have been uploaded to Dryad at DOI: https://doi.org/10.5061/dryad.8231nj1. Custom MATLAB scripts for generating and fitting perievent histograms to twitch and wake movement models can be found on github (https://github.com/jcdooley/Dooley_and_Blumberg_2018; archived at https://github.com/elifesciences-publications/Dooley_and_Blumberg_2018).

The following dataset was generated:

| Author(s) | Year | Dataset title | Dataset URL | Database and Identifier |
|---|---|---|---|---|
| Dooley J, Blumberg M | 2018 | Data from: Developmental | https://dx.doi.org/10.5061/dryad.8231nj1 | Dryad Digital Repository, 10.5061/ |

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
