## [Decision Letter]

Thank you for submitting your article "Developmental ‘awakening’ of primary motor cortex to the sensory consequences of movement" for consideration by *eLife*. Your article has been reviewed by two peer reviewers, including John Martin as Guest Reviewing Editor, and the evaluation has been overseen by Richard Ivry as the Senior Editor.

The reviewers have discussed the reviews with one another and the Reviewing Editor has drafted this decision to help you prepare a revised submission.

Summary:

This is an elegant study that provides novel insights into the development of sensorimotor circuitry and the role of sleep in the maturation of connectivity between the brain and the periphery. The study provides important mechanistic evidence, building on prior important findings from the senior author's lab, showing a developmental transition from responsiveness of motor cortex to sleep twitches, but not wake-related limb movements, to more selective responsiveness to wake-related limb movements. The study offers a novel general perspective on early corticospinal motor system development and is broadly impactful for brain systems development, through parallels with sensory system development.

Essential revisions:

1) The extent to which twitch- and wake-related events are the result of peripheral sensory signals in response to movement or internal signaling within the CNS/motor systems is difficult to distinguish with certainty. Moreover, evidence is presented of twitch-related activity in M1 leading movement onset (e.g., Figure 5, Figure 6, Figure 6—figure supplement 1) and in ECN that cannot be accounted for by receptive field activation, and complex changes in M1 after movement onset (i.e., up-tics in M1 firing after twitch onset; presence of prolonged activity). The presence of a peripheral receptive field can account for activity after movement onset, but it does not preclude internal motor signaling. Greater clarification of the timing and duration of activity in relation to movement onset and a more nuanced discussion of receptive field activation will strengthen the interpretations of the study.

2) Similar to #1, M1 neural activity (probably at all ages) likely reflects afferent input as much as motor output, but this does not mean it is a "sensory" structure. M1 is part of a sensory-motor control loop, albeit at a higher level than a reflex. This should be clarified because the authors overall hypothesis is tightly linked to this sensory-motor question.

3) Neural recording performed in unanaesthetized infant rats is extremely challenging and, although the authors are established leaders in this technique, the approach is potentially prone to technical issues, such as signal stability and artifacts. Most of the unit recordings are illustrated with raster plots, which do not allow assessment of recording quality, that may change instantaneously in relation to movement or across development. There are three points that should be considered: (A) It is important to illustrate the figures with raw MUA traces where possible, and provide spike waveforms for all individual 'representative' neurons shown; (B) It would strengthen the study if further data are provided showing that recordings obtained at P8 are as stable as at P12 (e.g. the same number of isolated neurons, signal-to-noise ratio, etc.); (C) Use of terminology such as 'putative single unit' instead of 'neuron' or 'unit' may be more appropriate.

4) Figure 9 is more of a summary than an overall model. The authors should consider using this figure to better illustrate their model for developmental transitions and post-twitch response suppression. Related more to a model then to this specific figure, the authors should take into consideration that GABA potentials are depolarizing early in development and there is an associated upregulation of KCC2 that coincides with a change in the chloride reversal potential and a hyperpolarizing GABA action. This is particularly relevant given the timing and the relatively abrupt changes in inhibitory control they observe.

5) An important issue that should be discussed is whether the effects observed are specific to M1. The authors mention "striking similarities between the activity profiles of S1 and M1 neurons to the same movements (Figure 5C, D)." Apropos to this comment, the authors could discuss (or have data addressing) the possibility that movement- or twitch-related cortical activity is more widespread, and is also present in cortical areas unrelated to movement control, such as other sensory cortices (visual, auditory). It would strengthen the study if nonspecific effects of 'global' changes in arousal levels (arising from subcortical neuromodulatory systems, for example), presumably associated with movements of any kind, can be distinguished from specific changes in local cortical areas that are directly involved in receiving local re-afference.

---

## [Author Response]

Essential revisions:1) The extent to which twitch- and wake-related events are the result of peripheral sensory signals in response to movement or internal signaling within the CNS/motor systems is difficult to distinguish with certainty. Moreover, evidence is presented of twitch-related activity in M1 leading movement onset (e.g., Figure 5, Figure 6, Figure 6—figure supplement 1) and in ECN that cannot be accounted for by receptive field activation, and complex changes in M1 after movement onset (i.e., up-tics in M1 firing after twitch onset; presence of prolonged activity). The presence of a peripheral receptive field can account for activity after movement onset, but it does not preclude internal motor signaling. Greater clarification of the timing and duration of activity in relation to movement onset and a more nuanced discussion of receptive field activation will strengthen the interpretations of the study.

We have added a paragraph in the Discussion to address the above points in greater depth (subsection “Developmental Changes in Reafferent Responses to Twitches”), and further explained our assertion about a lack of preceding activity in the Results (subsection “Rapid Developmental Onset of Sensory Responsiveness in M1 Neurons”).

The key point to explain here is that the “preceding” M1 activity is found exclusively for twitches. As we explain, this apparent increase results from the close temporal proximity of twitches in one muscle to twitches in other nearby muscles (see our description of this phenomenon in Blumberg et al., 2013). Because M1 has broad receptive fields at these ages, its activity increases to both the triggered muscle (i.e., the bicep here) and other muscles from which we are not recording.

2) Similar to #1, M1 neural activity (probably at all ages) likely reflects afferent input as much as motor output, but this does not mean it is a "sensory" structure. M1 is part of a sensory-motor control loop, albeit at a higher level than a reflex. This should be clarified because the authors overall hypothesis is tightly linked to this sensory-motor question.

We thank the reviewers for this comment, as we agree that referring to M1 as a “sensory structure” is potentially misleading. We have changed the sentence in the Discussion that refers to M1 as a sensory structure (first paragraph). We have also reviewed the entire paper to ensure that we are more careful when discussing this issue.

3) Neural recording performed in unanaesthetized infant rats is extremely challenging and, although the authors are established leaders in this technique, the approach is potentially prone to technical issues, such as signal stability and artifacts. Most of the unit recordings are illustrated with raster plots, which do not allow assessment of recording quality, that may change instantaneously in relation to movement or across development. There are three points that should be considered: (A) It is important to illustrate the figures with raw MUA traces where possible, and provide spike waveforms for all individual 'representative' neurons shown; (B) It would strengthen the study if further data are provided showing that recordings obtained at P8 are as stable as at P12 (e.g. the same number of isolated neurons, signal-to-noise ratio, etc.); (C) Use of terminology such as 'putative single unit' instead of 'neuron' or 'unit' may be more appropriate.

We agree that demonstrating recording quality and stability is important. To address this concern, we have added a panel to Figure 1—figure supplement 1 that demonstrates the stability of our recordings over the recording period as well as the similarity of the neural data across the ages examined. The left panels of this figure show 120 seconds of raw, filtered MUA at the beginning and end of data collection for representative P8 and a P12 rats. The right panels show the mean waveform sorted from the MUA at the start and end of data collection for that same neuron, along with an overlay of all waveforms included in the mean.

At the reviewer’s suggestion, we have also included insets showing the spike waveforms for the representative neurons shown in Figure 2, Figure 6—figure supplement 1, and Figure 8—figure supplement 1. The waveforms in Figure 8—figure supplement 1 show the average waveform in both the Pre and the Post injection periods, showing stability across the entire recording period for these longer experiments.

Finally, to further address this point as well as the request below to include more information about spike sorting methods, we have expanded the “Multiunit Activity and Spike Sorting” subsection of the Materials and methods.

4) Figure 9 is more of a summary than an overall model. The authors should consider using this figure to better illustrate their model for developmental transitions and post-twitch response suppression. Related more to a model then to this specific figure, the authors should take into consideration that GABA potentials are depolarizing early in development and there is an associated upregulation of KCC2 that coincides with a change in the chloride reversal potential and a hyperpolarizing GABA action. This is particularly relevant given the timing and the relatively abrupt changes in inhibitory control they observe.

In light of the reviewer’s concern, we thought it best to change the title of Figure 9 from “Model of Developmental Changes in Reafferent Processing” to simply “Developmental Changes in Reafferent Processing.”

With regard to the change in GABAergic neurons and the upregulation of KCC2, we agree that the timing of the change in twitch-responsiveness suggests a possible role for this mechanism. Without direct evidence we are reluctant to speculate in this figure; however, we have expanded the discussion of changes to excitatory-inhibitory balance and have included citations to papers addressing KCC2 and GABA across development (subsection “Developmental Changes in Reafferent Responses to Twitches”).

5) An important issue that should be discussed is whether the effects observed are specific to M1. The authors mention "striking similarities between the activity profiles of S1 and M1 neurons to the same movements (Figure 5C, D)." Apropos to this comment, the authors could discuss (or have data addressing) the possibility that movement- or twitch-related cortical activity is more widespread, and is also present in cortical areas unrelated to movement control, such as other sensory cortices (visual, auditory). It would strengthen the study if nonspecific effects of 'global' changes in arousal levels (arising from subcortical neuromodulatory systems, for example), presumably associated with movements of any kind, can be distinguished from specific changes in local cortical areas that are directly involved in receiving local re-afference.

The reviewer is correct to point out that, as written, the quoted sentence is misleading as it implies a more global response to twitches and wake movementsthroughout cortex. We have clarified this section to state that the similarities in the responses were restricted to the forelimb-responsive region of M1 and S1, and that these cortical areas were not responsive to twitches of other parts of the body (subsection “M1 Sensory Responses Originate in Deep Cortical Layers”). Although not discussed in the text, in collecting these data there were numerous animals where the electrode missed the forelimb representation of either M1 or S1, resulting in no responses to exafferent stimulation, twitches, or wake movements of the forelimb, providing further evidence of somatotopic restriction.

Notably, previous work in visual cortex has failed to identify any increase in activity in relation to myoclonic twitches (Mukherjee et al.,Journal of Neurophysiology, 2017). This suggests that the increases in neural firing after twitches observed in the present paper are restricted to sensorimotor cortex, and within these areas are to some extent topographically restricted.

We agree with the reviewer that cortical activity across these ages does vary in several important ways across behavioral state, most notably with respect to the overall baseline level of activity, which is higher in active sleep and lower during wake (see Figure 2, Figure 5, Figure 6—figure supplement 1, and Figure 8—figure supplement 1). However, because the focus of the present work and present analyses was on sensory activity, rather than global activity, we did not feel that this manuscript was the proper venue for analysis and discussion of these findings.